# ON-POLICY REINFORCEMENT FINE-TUNING WITH OFFLINE REWARD FOR MULTI-STEP EMBODIED PLANNING

## ABSTRACT

Embodied planning requires agents to make coherent multi-step decisions based on dynamic visual observations and natural language goals. While recent vision-language models (VLMs) excel at static perception tasks, they struggle in interactive environments. In this work, we introduce an on-policy reinforcement fine-tuning framework with offline rewards, that preserves the generalization benefits of RFT while addressing the challenges of sparse rewards and costly interaction, supported by solid theoretical guarantees. Our approach is evaluated on Embench, a recent benchmark for interactive embodied tasks, covering both in-domain and out-of-domain scenarios. Experimental results show that our method significantly outperforms models of similar or larger scale, including GPT-4o-mini and 70B+ open-source baselines, and exhibits strong generalization to unseen environments. This work highlights the potential of reinforcement-driven reasoning to advance multi-step planning in embodied AI.

## 1 INTRODUCTION

Embodied task planning serves as a cornerstone in hierarchical embodied AI systems(Shi et al., 2025b; Zhang et al., 2024a), where intelligent agents must not only perceive their environment but also reason and act within it to accomplish complex, real-world tasks(Duan et al., 2022). Unlike low-level controllers that govern precise trajectory execution(Zawalski et al., 2024; Kim et al., b), high-level planning is responsible for formulating coherent action sequences that translate complex instructions into manageable sub-tasks(Wu et al., 2023). While conventional language-based reasoning is confined to static, text-driven contexts(Lightman et al., 2023; Ye et al., 2025; Shao et al., 2024), embodied planning operates within dynamic, interactive environments that demand sequential decision-making across multiple steps. Despite recent

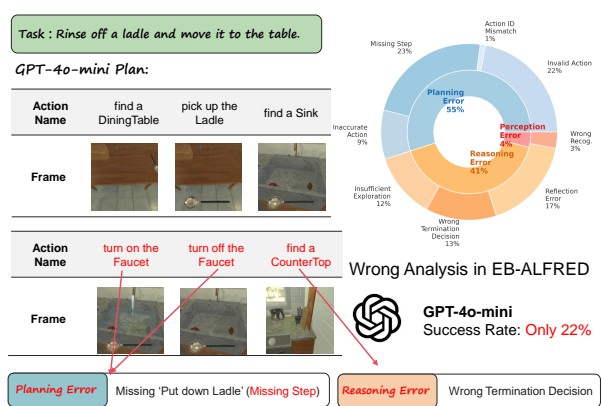

Figure 1: Failure case and error breakdown of GPT-4o-mini in the EB-ALFRED environment. **Left:** A representative task failure. **Right:** Distribution of failure types across EB-ALFRED tasks.

advancements in VLMs have demonstrated impressive capabilities in static understanding tasks(Zhang et al., 2024b), they exhibit substantial limitations when applied to **multi-step interactive** embodied planning. Empirical analyses in Figure1 reveal that even state-of-the-art VLMs, which excel in image captioning or visual question answering, struggle to maintain coherent and efficient decision sequences in dynamic environments(Yang et al., 2025). These shortcomings highlight a critical gap: effective planning in real-world embodied contexts imposes far greater demands on spatial reasoning, long-horizon coherence, and generalization capability than current VLM architectures can satisfy.

To address reasoning deficiencies, recent studies have demonstrated that reinforcement learning with verifiable rewards (RLVR) can substantially enhance the reasoning ability of large models and yield

stronger generalization across tasks compared to supervised fine-tuning (SFT) (Guo et al., 2025). Extensions of this paradigm into multimodal contexts have begun to emerge(Wang et al., 2025a), tackling tasks such as visual mathematics and diagram-based reasoning(Zhang et al., 2025a; Shen et al., 2025; Meng et al., 2025; Liu et al., 2025). However, applying such reasoning-enhancement techniques to embodied planning remains highly challenging due to the fundamental differences between embodied tasks and conventional reasoning benchmarks: **First, the shift from static, single-turn QA to interactive, multi-turn decision-making introduces continuous feedback loops** (Wang et al., 2025b) , **requiring agents to reason and act** while being grounded in dynamic environments. This also leads to sparse rewards as simulators typically provide only a final success/failure signal after many steps. **Second, acquiring reward signals through online interaction is prohibitively expensive**. During training, the model must interact with the environment while undergoing large-scale fine-tuning, which incurs heavy computational overhead on embodied benchmarks and becomes impractical when extending to real-world scenarios.

In this work, we bridge the gap by proposing an on-policy reinforcement fine-tuning framework with offline rewards for multi-turn embodied planning, which mitigates the challenges of costly interactions and sparse rewards while preserving the key advantages of RFT in generalization and reasoning capability. The key idea is to compute rewards by comparing model rollouts with expert trajectories, which are then used for downstream policy optimization via GRPO. Furthermore, We provide a theoretical analysis that clarifies how our offline reward differs from SFT and enjoys the benefit of RFT. Moreover, we prove that the policy induced by this reward is bounded relative to the policy learned with true online interaction, establishing that a lightweight offline approach can effectively substitute for costly online feedback. Recognizing the discrepancy between simplistic text-based simulations and the complexities of real-world physics, we conduct evaluations within Embench(Yang et al., 2025), an interactive embodied benchmark that faithfully captures environmental dynamics and agent-environment feedback loops. Experimental results demonstrate that our method significantly improves planning performance, yielding more efficient and context-aware action sequences. Moreover, our reinforcement-driven fine-tuning exhibits strong generalization across unseen tasks and environments, underscoring its potential for practical deployment in real-world embodied AI applications.

Our contributions are as follows:

- We are the first to apply reinforcement fine-tuning to optimize a vision-language model for embodied task planning, significantly improving the model's ability to perform coherent multi-step reasoning and decision-making in dynamic environments.
- We design an on-policy RFT framework with offline rewards that preserves the generalization benefits of RFT while addressing the challenges of costly simulation and sparse rewards. We further provide a theoretical analysis showing its distinction from SFT and proving that the policy induced by offline rewards admits a bounded gap to that learned with true online interaction.
- We conduct extensive evaluation on Embench, an interactive benchmark for embodied AI, showing that our model not only outperforms comparable-scale models but also surpasses GPT-4o-mini and open-source models with more than 70B parameters. It further demonstrates strong generalization to unseen domains, validating the generality of reinforcement-based adaptation.

## 2 METHODOLOGY

### 2.1 PROBLEM DEFINITION

We formulate embodied task planning as a multi-turn, partially observable decision-making process, where the agent interacts with an environment through sequential actions based on visual observations. At each time step $t$, the agent receives an observation $o_t \in \mathcal{O}$ and executes an action $a_t \in \mathcal{A}$, forming a history $h_t = \{o_0, a_0, o_1, ..., o_t\}$.

Given a task instruction $g \in \mathcal{G}$ described by a natural language command $L$, the task is associated with a set of binary goal-checking conditions $\mathcal{C}(g) = \{c_1, ..., c_k\}$ that must all be satisfied for the task to be considered successful. The agent generates a trajectory $e = (g, o_0, a_0, o_1, ..., o_n, a_n)$, and the environment reward is defined as $r(e) = \mathbb{I}\big[\mathcal{C}(g) \subseteq \{\text{True}\}\big]$, where $\mathbb{I}[\cdot]$ is the indicator function.

We parameterize the policy $\pi_\theta$ using a vision-language model (VLM), which outputs an action distribution conditioned on the observation $o_t$, history $h_t$, instruction $L$, and a fixed prompt $P$:

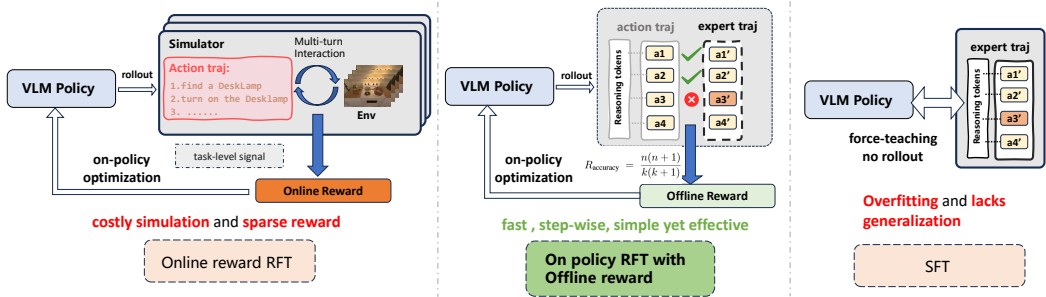

Figure 2: Comparison of our on-policy RFT with offline rewards against other methods. Our approach avoids costly simulation overhead and sparse reward issues, while achieving stronger generalization through reinforcement fine-tuning.

$$a_{t+1} \sim \pi_\theta(\cdot \mid o_t, h_t, L, P). \tag{1}$$

Our objective is to optimize $\theta$ such that the expected task success rate increases:

$$\max_\theta \mathbb{E}_{e \sim \pi_\theta} [r(e)]. \tag{2}$$

## 2.2 On-policy RFT for Embodied Planning with Offline Reward

While reinforcement fine-tuning (RFT) has been shown to retain prior knowledge and elicit reasoning compared to supervised fine-tuning (SFT), its application to embodied planning poses unique challenges. First, acquiring rewards via online interaction—where the agent executes sampled trajectories in simulators to collect feedback—is prohibitively expensive: each rollout requires environment resets, step-wise rendering, and physics simulation, which scale poorly in LLM post-training and impractical for real-world scenarios. Second, embodied simulators expose only an episodic success/failure signal, which poses the sparse reward problem for multi-turn embodied planning.

To address these limitations, we adopt an *offline reward optimization* approach that avoids online execution. Instead of collecting interactive feedback, we compute reward by comparing model-generated plans to expert trajectories. This design not only circumvents costly simulator interaction but also alleviates reward sparsity while retaining the generalization advantages over SFT. We will present the methodological overview in this section and provide deeper analysis in Section 2.3.

**Offline Expert Trajectory Construction.**    We construct our offline expert dataset based on the ALFRED benchmark (Shridhar et al., 2020), which provides complete ground-truth trajectories for household tasks in simulated environments. Each expert trajectory $e = (g, o_0, a_0, o_1, a_1, ..., o_k, a_k)$ is decomposed into $k$ training samples, specifically, for each step $n \in [1, k]$, we build an input prompt $L_n$ containing the task goal $g$ and the preceding action history $a_{0:n-1}$. The corresponding visual observation $o_n$ is taken from the $n$-th step, and the target response $\hat{a}_{n:} = \{a_n, ..., a_k\}$ includes all remaining actions. Applying this decomposition to the ALFRED dataset yields 43,898 training samples for reinforcement fine-tuning.

**Reward Function with Expert Trajectory.**    We instantiate an *offline* reward from expert trajectories: given the model's rollout and its expert action sequence to score the reward for policy optimization.

Let $\hat{a} = \{a_1, \ldots, a_k\}$ be the predicted sequence and $a^* = \{a_1^*, \ldots, a_k^*\}$ the expert reference. Define $n = \max\{i : a_j = a_j^* \; \forall j \leq i\}$ as the length of the longest correct prefix. We use a smooth, long-horizon–aware shaping term:

$$R_{\text{accuracy}} = \frac{n(n+1)}{k(k+1)}, \tag{3}$$

which emphasizes longer consecutive matches and yields stable gradients. Notably, although the quadratic prefix-based reward is **simple**, we find it **highly effective**, surpassing more complex alternatives. We further analyze its impact in section 3.3.

Complementary to correctness reward, we incorporate a format reward as an auxiliary signal. Prior studies have shown that structural regularization improves stability and prevents degenerate outputs, Following prior reinforcement fine-tuning practices (Meng et al., 2025; Tan et al., 2025),we set the maximum format reward to 0.5 and the accuracy reward to 1.0, yielding the final composite reward. We provide the detailed format reward components in the Appendix F.2.

$$R(\text{response, answer}) = R_{\text{accuracy}}(\text{response, answer}) + R_{\text{format}}(\text{response}) . \tag{4}$$

**Training Pipeline.** For the overall pipeline, we first perform supervised fine-tuning to establish structured planning priors, and then apply reinforcement fine-tuning as the main part, with the offline reward to strengthen multi-step reasoning. Concretely, the model generates candidate rollouts, which are scored against expert trajectories by our reward function; policy parameters are then updated using GRPO with an additional data filtering strategy to ensure stable optimization.

(1) Supervised Fine-tuning (SFT). We initialize the vision-language model by distilling expert-style trajectories, training via maximum likelihood to align with commonsense patterns and structured conventions. This provides a strong initialization for downstream reinforcement learning.

(2) On-policy Reinforcement Fine-tuning (RFT) with GRPO. Building on the offline reward signals, we adopt Group Relative Policy Optimization (GRPO) (Shao et al., 2024) to optimize the policy. For each prompt, the model samples multiple candidate responses, each scored by the reward function. Relative advantages are then computed within the group, and gradients encourage high-reward responses while regularizing against deviation from a reference policy:

$$\mathcal{J}(\theta) = \mathbb{E}_{x \sim \mathcal{D}} \, \mathbb{E}_{\{y_i\} \sim \pi_\theta} \left[ \frac{1}{G} \sum_{i=1}^{G} \left( \text{clip} \left( \frac{\pi_\theta(y_i \mid x)}{\pi_{\text{old}}(y_i \mid x)}, \, 1 - \epsilon, \, 1 + \epsilon \right) \cdot A_i - \beta \cdot \mathcal{D}_{\text{KL}}(\pi_\theta \| \pi_{\text{ref}}) \right) \right] . \tag{5}$$

(3) Data Filtering. To ensure informative and stable gradients, we incorporate an online filtering strategy during RFT, inspired by PRIME (Cui et al., 2025) and MM-Eureka (Meng et al., 2025). Prompt groups with too few or too many perfect-reward responses are discarded, maintaining a balanced learning signal and stabilizing training.

**Conclusion.** In this section, we introduced our RFT framework for multi-step embodied planning. The core idea is to compute **offline** rewards by prefix-based comparison with expert trajectories, and to optimize the policy through **on-policy** GRPO. In the next section, we provide a deeper theoretical analysis of these properties, clarifying the effectiveness of our approach and its distinctions from existing methods.

## 2.3 DEEPER ANALYSIS

We conduct a deeper analysis from two perspectives: (1) the distinction and advantages of offline reward RFT over SFT, given that both rely on offline ground-truth datasets for supervision (Section 2.3.1); and (2) the validity of using offline trajectories as a proxy for real environmental feedback, with an upper bound on the theoretical gap between the two methods (Section 2.3.2)

### 2.3.1 SFT VS. ONLINE/OFFLINE REWARD RFT: FORMAL COMPARISON.

We present the formulations of SFT, offline reward RFT, and online reward RFT as follows.

**SFT (token-level cross-entropy on all labels).** Here every output token (reasoning or action) is directly supervised, enforcing strict imitation:

$$\mathcal{L}_{\text{SFT}}(\pi) = - \mathbb{E}_{x \sim D, \, y \sim \pi_\beta(\cdot | x)} \left[ \sum_{t=1}^{k} \left( \underbrace{\log \pi(y_r \mid x)}_{\text{reasoning tokens}} + \underbrace{\log \pi(y_a \mid x)}_{\text{action tokens}} \right) \right] . \tag{6}$$

While effective, SFT limits reasoning diversity and generalization, whereas RFT alleviates these limitations. The following two formulations present the details of RFT, whose main difference lies in the reward source (online vs. offline).

**Online reward RFT (policy-gradient with environment feedback).**

$$\mathcal{J}_{\text{RFT}}^{\text{on}}(\pi) = \mathbb{E}_{x \sim D, \ y \sim \pi(\cdot \mid x)} \Big[ \underbrace{A_{\text{env}}(x, y_a)}_{\text{from env. reward;}} \log \pi(y \mid x) \Big] - \beta \, D_{\text{KL}}(\pi \,\|\, \pi_{\text{ref}}). \quad (7)$$

**Offline reward RFT (ours; on-policy sampling with expert-derived reward).**

$$\mathcal{J}_{\text{RFT}}^{\text{off}}(\pi) = \mathbb{E}_{x \sim D, \ y \sim \pi(\cdot \mid x)} \Big[ \underbrace{A_{\text{off}}(x, y_a; y_E)}_{\text{from expert-comparison}} \log \pi(y \mid x) \Big] - \beta \, D_{\text{KL}}(\pi \,\|\, \pi_{\text{ref}}), \quad (8)$$

where $y_E$ is the expert trajectory and $A_{\text{off}}$ is computed from an offline *reward model* $r_{\text{off}}(x, y; y_E)$ (e.g., prefix-accuracy) via a group-wise standardization to form advantages in our settings.

Although both SFT and our offline reward RFT rely on offline datasets for supervision, the formulations reveal two key differences:

*Outcome-level vs. token-level supervision (Difference 1).* In equation 6, the learning signal is per-token and covers *all* output tokens (including intermediate reasoning). While this ensures strict imitation, it also constrains the diversity of reasoning trajectories and increases the risk of overfitting to specific paths. In contrast, in equation 7, equation 8, the scalar advantage $A(\cdot)$ depends only on action tokens $y_a$, as a result, reasoning tokens are shaped indirectly through sequence-level weighting, preserving flexibility and encouraging diverse reasoning pathways.

*Sampling distribution & negative examples (Difference 2).* Prior work (Shenfeld et al., 2025) has shown that *on-policy* sampling and *negative examples* are the key reasons why RFT achieves better generalization and avoids forgetting compared to SFT. Both equation 7 and equation 8 optimize under the current policy distribution $y \sim \pi(\cdot \mid x)$, unlike SFT's fixed $\pi_\beta$. Also, RFT allows low-quality samples to receive negative advantages($A < 0$). In conclusion, our on-policy offline reward RFT inherits these two properties, despite relying on an expert-derived reward source, thereby aligning its training dynamics with online reward RFT.

The above differences show our offline reward RFT inherits the key properties of RFT–stronger generalization compared to SFT–which is further corroborated by the experiments in Section 3.2.

### 2.3.2 OPTIMAL-POLICY DISTANCE UPPER BOUND.

In our approach, offline rewards replace real environmental feedback, raising the natural question of whether such a substitution is valid. In this section, we provide a theoretical upper bound on the distance between the policies induced by the two reward formulations:

For a fixed prompt $x$, consider the KL-regularized objective

$$\mathcal{J}_r(\pi \mid x) = \mathbb{E}_{y \sim \pi(\cdot \mid x)}\big[r(y)\big] - \beta_{\text{eff}} \, D_{\text{KL}}\big(\pi(\cdot \mid x) \,\|\, \pi_{\text{ref}}(\cdot \mid x)\big), \quad (9)$$

where $\beta_{\text{eff}} > 0$ and $\pi_{\text{ref}}(\cdot \mid x)$ has full support. It is standard that the maximizer has the Gibbs form

$$\pi_r^\star(y \mid x) = \frac{\pi_{\text{ref}}(y \mid x) \, \exp\big(r(y)/\beta_{\text{eff}}\big)}{\sum_{y'} \pi_{\text{ref}}(y' \mid x) \, \exp\big(r(y')/\beta_{\text{eff}}\big)}. \quad (10)$$

Let $y_E$ be a fixed expert trajectory of length $k$. Define the *expert-comparison* reward $r_{\text{exp}}$ via the longest correct prefix, While the *environment* success reward is not tied to a single trajectory but to the whole success set $\mathcal{Y}_{\text{succ}}(x)$.

$$r_{\text{exp}}(y) = \frac{n(y)\big(n(y)+1\big)}{k(k+1)}, \qquad r_{\text{env}}(y) = \mathbb{I}\{ y \in \mathcal{Y}_{\text{succ}}(x) \} \quad (11)$$

Let the *minimal expert-prefix among all successful trajectories* be $n_{\text{min}}(x) = \min_{y \in \mathcal{Y}_{\text{succ}}(x)} n(y)$. Then the smallest expert-comparison reward among successful trajectories is

$$\min_{y \in \mathcal{Y}_{\text{succ}}(x)} r_{\text{exp}}(y) = \frac{n_{\text{min}}(x)\big(n_{\text{min}}(x) + 1\big)}{k(k + 1)}. \quad (12)$$

We upper bound the pointwise gap:

$$\delta_\star(x) = \big\|r_{\exp} - r_{\mathrm{env}}\big\|_\infty \;\leq\; \max\left\{ 1 - \frac{n_{\min}(x)\big(n_{\min}(x)+1\big)}{k(k+1)}, \; \frac{k-1}{k+1} \right\}. \qquad (13)$$

The first term controls the max discrepancy *among successes* ($y \in \mathcal{Y}_{\mathrm{succ}}$), while the second term controls the max discrepancy *among failures* ($y \notin \mathcal{Y}_{\mathrm{succ}}$), using $\max_{n \leq k-1} \frac{n(n+1)}{k(k+1)} = \frac{k-1}{k+1}$.

**Lemma 1 (Softmax stability).** Let $r_1, r_2 : \mathcal{Y} \to \mathbb{R}$ be bounded rewards with $\|r_1 - r_2\|_\infty \leq \delta$. Let $\pi_i^\star$ be the maximizer of equation 9 with $r = r_i$ (thus of the form equation 10). Then

$$D_{\mathrm{KL}}\big(\pi_1^\star \,\|\, \pi_2^\star\big) \;\leq\; \frac{2\delta}{\beta_{\mathrm{eff}}}, \qquad D_{\mathrm{KL}}\big(\pi_2^\star \,\|\, \pi_1^\star\big) \;\leq\; \frac{2\delta}{\beta_{\mathrm{eff}}}. \qquad (14)$$

Detailed proof will be given in Appendix D

**Proposition 1 (Expert-vs-Environment optimal policies are bounded).** Let $\pi_{\exp}^\star$ and $\pi_{\mathrm{env}}^\star$ be the maximizers of equation 9 with $r = r_{\exp}$ and $r = r_{\mathrm{env}}$, respectively. Then, with $\delta_\star(x)$ in equation 13, we get the policy distance by Pinsker's inequality,

$$D_{\mathrm{KL}}\big(\pi_{\exp}^\star \,\|\, \pi_{\mathrm{env}}^\star\big) \;\leq\; \frac{2\,\delta_\star(x)}{\beta_{\mathrm{eff}}}, \qquad \big\|\pi_{\exp}^\star - \pi_{\mathrm{env}}^\star\big\|_{\mathrm{TV}} \;\leq\; \sqrt{\tfrac{1}{2} D_{\mathrm{KL}}\big(\pi_{\exp}^\star \,\|\, \pi_{\mathrm{env}}^\star\big)} \;\leq\; \sqrt{\frac{\delta_\star(x)}{\beta_{\mathrm{eff}}}}. \qquad (15)$$

Inequalities equation 15 establish that the optimal policy under the offline *expert-comparison* reward remains provably close to the environment-optimal policy. Importantly, this does not imply that the sparse environment reward is always superior: when a trajectory ultimately fails, the environment reward collapses to zero, whereas our expert-based reward can still assign a positive signal proportional to prefix similarity. This alleviates **reward sparsity problem** in such multi-turn settings while the bounded discrepancy guarantees that optimization under the offline reward remains directionally aligned with the online environment objective.

## 3 EXPERIMENTS

We conduct a series of experiments to evaluate the effectiveness of our proposed reinforcement fine-tuning (RFT) framework for multi-step embodied planning. Specifically, we aim to answer the following key questions:

   **(Q1)** *How well does our method perform in interactive benchmarks for multi-step embodied task planning?* (Section 3.1)

   **(Q2)** *Is reinforcement fine-tuning necessary and uniquely beneficial, especially compared to supervised fine-tuning?* (Section 3.2)

   **(Q3)** *Is our prefix-based offline reward design truly effective, for example when compared to more complex reward formulations?* (Section 3.3)

### 3.1 EXPERIMENT RESULTS IN EMBENCH (Q1)

#### 3.1.1 EXPERIMENTAL SETTINGS

**Benchmark**   Most prior works in embodied planning reduce evaluation to static visual question answering, which fails to capture the interactive and sequential nature of real-world decision-making. To address this gap, we adopt **Embench**(Yang et al., 2025), a benchmark designed for evaluating multimodal agents in dynamic, interactive environments.

Embench provides a unified framework across four embodied settings and supports over 1,100 tasks involving manipulation, navigation, and spatial reasoning. We evaluate on two environments: **EB-ALFRED**, built on ALFRED(Shridhar et al., 2020) and AI2-THOR(Kolve et al., 2017), and **EB-Habitat**, based on Habitat 2.0's rearrangement tasks(Savva et al., 2019). The benchmark organizes tasks into different subsets. Among them, the *Base* set forms the core task pool, while the *Common*

| Model | EB-ALFRED (Seen) | | | | | | EB-Habitat (Unseen) | | | | | |
|---|---|---|---|---|---|---|---|---|---|---|---|---|
| | Avg | Base | Com | Cplx | Visual | Spatial | Avg | Base | Com | Cplx | Visual | Spatial |
| **Closed-Source MLLMs** | | | | | | | | | | | | |
| Claude-3.5-Sonnet | 65.2 | 70 | 62 | 72 | 62 | 60 | 70.4 | 96 | 68 | 74 | 74 | 40 |
| Gemini-2.0-flash | 50.8 | 58 | 58 | 50 | 46 | 42 | 38.4 | 76 | 30 | 30 | 30 | 26 |
| GPT-4o | 54.8 | 62 | 52 | 68 | 44 | 48 | 53.6 | 82 | 34 | 62 | 58 | 32 |
| GPT-4o-mini | 26.4 | 32 | 24 | 32 | 20 | 24 | 36.8 | 68 | 38 | 28 | 28 | 22 |
| **Open-Source MLLMs** | | | | | | | | | | | | |
| LLaMA-3.2-90B | 35.2 | 38 | 34 | 44 | 28 | 32 | 45.6 | 94 | 24 | 50 | 32 | 28 |
| LLaMA-3.2-11B | 15.2 | 24 | 8 | 16 | 22 | 6 | 26.8 | 62 | 16 | 24 | 14 | 18 |
| Qwen2.5-VL-72B | 40.8 | 50 | 42 | 42 | 36 | 34 | 41.2 | 72 | 28 | 42 | 40 | 24 |
| Qwen2.5-VL-7B | 2.0 | 4 | 2 | 2 | 2 | 0 | 14 | 38 | 4 | 12 | 4 | 12 |
| InternVL2.5-78B | 36.8 | 38 | 34 | 42 | 34 | 36 | 53.2 | 80 | 42 | 56 | 58 | 30 |
| InternVL2.5-8B | 3.6 | 2 | 0 | 12 | 0 | 4 | 19.6 | 48 | 6 | 16 | 10 | 18 |
| **Open-Source Reasoning MLLMs** | | | | | | | | | | | | |
| R1-VL-7B | 2 | 2 | 2 | 6 | 0 | 0 | 8.4 | 24 | 0 | 4 | 6 | 8 |
| MM-Eureka-Qwen-7B | 3.2 | 6 | 4 | 4 | 2 | 0 | 19.2 | 40 | 16 | 14 | 10 | 16 |
| **Open-Source Embodied MLLMs** | | | | | | | | | | | | |
| RoboBrain | 0.4 | 2 | 0 | 0 | 0 | 0 | 17.6 | 38 | 6 | 18 | 8 | 18 |
| Tapa | 0.0 | 0 | 0 | 0 | 0 | 0 | 0.0 | 0 | 0 | 0 | 0 | 0 |
| **Open-Source Embodied + Reasoning MLLMs** | | | | | | | | | | | | |
| Ours(base) | 2.0 | 4 | 2 | 2 | 2 | 0 | 14 | 38 | 4 | 12 | 4 | 12 |
| Ours(SFT) | 22 | 34 | 22 | 24 | 12 | 18 | 13.6 | 34 | 2 | 10 | 10 | 12 |
| Ours(SFT+RFT) | **49.2** | 60 | 60 | 48 | 38 | 40 | **22.4** | 56 | 8 | 18 | 16 | 14 |

Table 1: Side-by-side comparison: left EB-ALFRED (Seen) vs. right EB-Habitat (Unseen). **Abbreviations:** Com = Common, Cplx = Complex

*Sense*, *Complex Instruction*, *Spatial Awareness*, *Visual Appearance* are constructed via prompt-level augmentation that increases reasoning or perception difficulty, such as adding commonsense constraints or syntactic complexity. Notably, our RL fine-tuning is conducted solely on the *Base* set without any prompt augmentation, demonstrating its ability to generalize beyond the training distribution.

All models generate step-by-step plans from egocentric inputs and execute them in simulation. Since our training data is collected from the ALFRED , EB-Habitat serves as an fully **out-of-domain** setting for generalization evaluation. More details are provided in Appendix G.

**Baselines** We compare our method against a range of baselines, including: (1) proprietary models such as Claude-3.5-Sonnet(cla), Gemini-2.0-flash(gem), GPT-4o(gpt, a), and GPT-4o-mini(gpt, b); (2) open-source general VLMs like LLaMA-3.2-Vision-11B(lla), Qwen2.5-VL-7B(Bai et al., 2025), and InternVL2.5-8B(Chen et al., 2024); (3) reasoning-oriented models such as MM-Eureka(Meng et al., 2025) and R1-VL(Zhang et al., 2025a); and (4) embodied VLMs including RoboBrain(Ji et al.) and TAPA(Wu et al., 2023). For evaluation, we convert visual inputs into text for TAPA due to its lack of vision capabilities. Further details on each baseline are provided in Appendix G, while some very recent models like Shi et al. (2025a); Zhang et al. (2025b) are excluded due to unavailability of open-source implementations and reproducibility challenges.

**Evaluation Metrics** We follow the original Embench (Yang et al., 2025) to use **task success rate** as the primary evaluation metric. A task is marked as successful only if all predefined goal-checking conditions are satisfied at the end of execution.

To support multi-turn planning, Embench adopts an iterative evaluation protocol where the model generates a new action sequence based on the latest observation at each round. The environment executes the actions and returns updated states until task success or step limit is reached.

### 3.1.2 MAIN RESULTS IN EMBENCH

**In-Domain Results** We conduct comprehensive in-domain evaluations on the EB-ALFRED environment. As shown in Table 1, our proposed model achieves a task success rate of 49.2%, approaching

Gemini-2.0-flash (50.8%) and surpassing all other open-source models including Qwen2.5-VL-72B(40.8%) and Llama-3.2-90B(35.2%).

Several key observations emerge from the results: (1) Our methods and training pipeline lead to consistent performance gains in embodied task planning for both base and other advanced tasks. (2) Existing open-source reasoning models and embodied VLMs perform poorly in Embench. While reasoning models produce verbose intermediate steps, they struggle to execute correct action sequences. Similarly, embodied VLMs lack the generalization ability to transfer to Embench tasks.

| Variant | EB-ALFRED (Seen) | | EB-Habitat (Unseen) | |
|---|---|---|---|---|
| | Avg | Base | Avg | Base |
| Base | 2 | 4 | 14 | 38 |
| SFT only | 22 | 34 | 13.6 | 30 |
| RFT only | 14.4 | 22 | 17.6 | 40 |
| RFT → SFT | 33.2 | 40 | 11.4 | 30 |
| SFT → SFT | 37.6 | 50 | 11.6 | 22 |
| SFT → RFT (ours) | **49.2** | **60** | **22.4** | **56** |

(a) Ablation study on training stages in EB-ALFRED and EB-Habitat.

**Out-of-Domain Results** To evaluate generalization, we tested our models in the EB-Habitat environment, which differs from ALFRED in terms of scenes, objects, action space, and task types. As shown in the right part of Table 1, our method exhibits strong out-of-domain performance, outperforming all baseline models of similar 7B size, including general-purpose, reasoning-augmented, and embodied VLMs.The

| Model | Overall Acc | SpatialMap | MazeNav | SpatialGrid |
|---|---|---|---|---|
| Base | 0.475 | 0.696 | 0.256 | 0.542 |
| SFT only | 0.488 | 0.682 | **0.328** | 0.524 |
| SFT+RFT | **0.503** | **0.748** | 0.260 | **0.605** |

(b) Visual reasoning accuracy on spatial VQA subsets.

Table 2: RFT Generalization Experiment

result highlight Reinforcement fine-tuning leads to substantial improvements even in completely unseen environments.

## 3.2 RFT Generalizes While SFT Overfits (Q2)

**Is Reinforcement Fine-Tuning Necessary?** A key question is whether the performance gain of GRPO-based reinforcement fine-tuning (RFT) stems from the optimization process itself, or merely from exposure to additional trajectory data. To investigate this, we compare five training strategies: (1) **Base**: the original Qwen2.5-VL-7B model without any tuning; (2) **SFT only**: supervised fine-tuning (SFT) on distilled trajectories; (3) **RFT→SFT**: first applying RFT, then re-align with SFT; (4) **SFT→SFT**: conducting SFT , followed by additional SFT using the same trajectories during RFT. This variant isolates the effect of data exposure from optimization. and (5) **SFT→RFT (ours)**: our proposed pipeline with SFT followed by GRPO-based RFT.

As shown in Table 2a, our **SFT → RFT** pipeline achieves the best performance across both seen and unseen environments. While **SFT → SFT** brings moderate gains over **SFT only** on seen tasks, it surprisingly degrades performance in unseen domains—exposing the limitations of supervised fine-tuning. In contrast, our **SFT → RFT** approach not only boosts more in-domain accuracy but also enhances generalization, confirming the necessity of offline reward-driven optimization beyond simple trajectory exposure.

**Does RFT Overfit to Embodied Benchmarks?** To further evaluate the generalization capability of RFT, we assess whether fine-tuning on Embench harms the model's performance on its original training domains besides embodied task planning. Specifically, we evaluate on **SpatialEval** (Wang et al., 2024a), a benchmark designed to assess general spatial understanding across three diverse tasks: spatial maps, maze navigation, and spatial grids.

As shown in Table 2b, the SFT-RFT model not only avoids degradation on general spatial reasoning tasks but also improves performance on spatial map and spatial grid tasks. This indicates that our reinforcement-based fine-tuning pipeline promotes structured reasoning without overfitting to the embodied benchmark. The structured action plans and reward-aligned outputs learned through RFT appear to benefit broader visuospatial understanding.

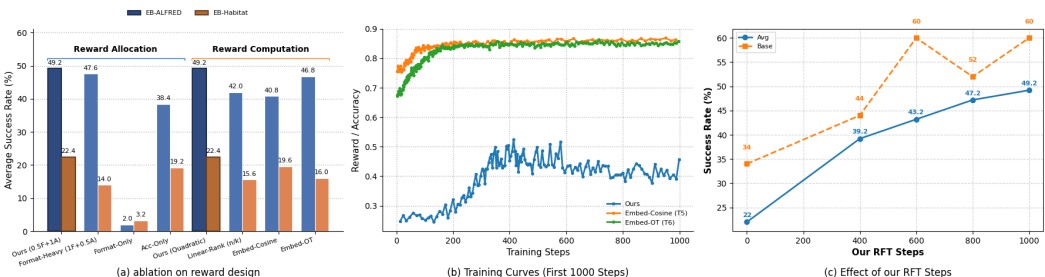

Figure 3: Ablation and training analyses of our reward design.(a) Comparison of different reward allocation and computation strategies.(b) Training curves of three method. (c) Effect of RFT steps on Avg and Base success rates.(evaluation)

## 3.3 ABLATION STUDY ON REWARD DESIGN (Q3)

To validate the effectiveness of our prefix-based reward, we conduct a series of ablation studies focusing on reward allocation, reward computation, and training dynamics, as illustrated in Figure 3.

**(a) Reward allocation.** We compare different allocations between accuracy and format signals, including our design (`1Accuracy+0.5Format`), `Format-Heavy (1Format+0.5Accuracy)`, `Format-Only`, and `Accuracy-Only`. As shown in Figure 3(a, left), our allocation achieves the best performance, followed by the format-heavy variant. `Format-Only` fails to provide meaningful gains, while `Acc-Only` underperforms ours, confirming that format guidance plays a necessary role.

**(b) Reward computation.** We further compare different reward formulations: our quadratic prefix curve $n(n + 1)/k(k + 1)$, a linear prefix variant $n/k$, and two embedding-based methods (`Embed-Cosine` and `Embed-OT`). In `Embed-Cosine`, both the rollout and expert trajectory are encoded using the Qwen3-Embedding-0.8B model, and the cosine similarity between their embeddings is taken as the reward. In `Embed-OT`, the same embeddings are compared via an OT(optimal transport) distance computed with the Sinkhorn algorithm(Cuturi, 2013).

Results in Figure 3(a, right) show that our simple prefix reward consistently outperforms embedding-based similarities. As highlighted in Figure 3(b), embedding-based rewards quickly saturate within 200 steps, yielding high training scores but weak test performance. In contrast, our prefix reward produces lower and noisier training signals, yet, as shown in Figure 3(c), test performance steadily improves. We attribute this to its simplicity: the prefix-based design introduces greater learning difficulty, which acts as an implicit regularizer, preventing the model from exploiting spurious correlations and instead encouraging gradual long-horizon alignment with expert trajectories for more robust policy optimization.

**(c) Effect of training steps.** Finally, we examine how success rates evolve with different RFT steps. Figure 3(c) shows that the *Base* subset saturates around 500 steps, but continued training yields steady gains in other subsets (*Visual*, *Spatial*, *Complex*), driving the overall *Avg* upward from 22.0 to 49.2. This emergent transfer is particularly noteworthy given that our RFT training data exclusively contains *Base*-type tasks, highlighting the generalization capability induced by our reinforcement optimization process.

## 4 CONCLUSION

In this work, we addressed the challenge of multi-step embodied planning by introducing an on-policy RFT framework with offline rewards. Our approach avoid costly simulator interaction and sparse reward while preserving the generalization benefits of RFT. Supported by theoretical analysis and extensive experiments on Embench, we demonstrated that this method significantly improves reasoning and decision-making capabilities. More broadly, our results highlight the potential of offline reward–driven reinforcement tuning as a scalable paradigm for advancing embodied AI and other multi-turn agentic RL settings, enabling vision-language models to perform coherent long-horizon planning and paving the way toward practical deployment in real-world applications.

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

## A  APPENDIX CONTENTS

## B  USE OF LLMS

Large language models (LLMs) were employed to provide assistance with language refinement during the preparation of this paper. We emphasize that the LLMs were used only as writing aids; all conceptual contributions—including research problem formulation, methodological design, experimental analysis, and interpretation of findings—are the sole work of the authors.

## C  RELATED WORK

### C.1  EMBODIED TASK PLANNING

Embodied task planning focuses on decomposing high-level natural language instructions into executable sequences of sub-tasks, enabling agents to perform complex behaviors in interactive environments. With the emergence of large language and vision-language models(Xi et al., 2025; Xu et al., 2024), researchers have explored using pretrained LLMs or VLMs to generate plans from textual and visual observations, typically relying on carefully crafted prompts(Shin et al.; Rana et al.; Hu et al.; Kim et al., a; Singh et al.; Fu et al., 2024) or auxiliary tools(Rana et al.; Ahn et al.; Silver et al.) to provide necessary planning cues. While simple and data-efficient, such methods often struggle with spatial grounding and temporal coherence in visually rich environments. Advanced methods have tried to fine-tune LLMs or VLMs to improve planning performance. Several works have employed supervised fine-tuning pipelines(Wu et al., 2023; Chen et al.; Ji et al.), while others adopt preference optimization methods(Wang et al.; Song et al., 2024) such as Direct Preference Optimization (DPO)(Rafailov et al., 2023) to better align model behavior with expert planning preferences.

Despite these advances, most existing methods operate in static or offline settings, where plans are generated without actual interaction with the environment. In this work, we address this limitation by evaluating our model in interactive environments(Yang et al., 2025).while a recent work (Shi et al., 2025a) also adopts Embench, their focus lies in data augmentation and SFT, whereas we study reinforcement fine-tuning and propose an on-policy RFT with offline rewards.

### C.2  VISION-LANGUAGE MODEL REASONING

Reasoning in vision-language models (VLMs) involves drawing inferences from both textual and visual inputs, often requiring spatial, temporal, or causal understanding(Wang et al., 2025a; 2024b). A common approach is Chain-of-Thought (CoT) prompting(Wei et al., 2022), where the model generates intermediate steps to clarify its reasoning. In multimodal settings, Multimodal Chain-of-Thought (MCoT) extends this idea by integrating visual inputs like images and videos into the reasoning process(Zhang et al., 2023; Mondal et al., 2024; Mitra et al., 2024).

More recently, R1-style reinforcement learning(Guo et al., 2025; Shao et al., 2024) has emerged as an effective framework for enhancing reasoning capabilities. These methods optimize reasoning quality through reward-guided learning, enabling models to self-correct and generate more detailed reasoning processes. Originally developed for text-based reasoning, R1 approaches have since been extended to multimodal domains, including image-based QA(Liu et al., 2025; Shen et al., 2025; Zhang et al.,

2025a), visual math problems(Meng et al., 2025; Huang et al., 2025; Team et al., 2025), and video reasoning(Li et al., 2025). In the context of embodied AI, some early studies(Zhao et al., 2025; Zhang et al., 2025b; Tan et al., 2025) have applied R1-based training to question answering tasks, however, they primarily focus on short-horizon QA tasks.In contrast, our work is the first to adopt R1-style reinforcement fine-tuning for long-horizon embodied planning, addressing the challenges of sparse and costly online rewards while preserving the generalization benefits of RFT for structured multi-step decision-making.

# D   PROOF OF LEMMA 1 (SOFTMAX STABILITY)

**Lemma 1 (Softmax stability).** Let $r_1, r_2 : \mathcal{Y} \to \mathbb{R}$ be bounded rewards on a finite (or countable) outcome space $\mathcal{Y}$, and assume a reference policy $\pi_{\mathrm{ref}}(\cdot \mid x)$ with full support. Fix $\beta_{\mathrm{eff}} > 0$ and define, for $i \in \{1, 2\}$,

$$\pi_i^\star(y \mid x) = \frac{\pi_{\mathrm{ref}}(y \mid x) \exp\big(r_i(y)/\beta_{\mathrm{eff}}\big)}{\sum_{y'} \pi_{\mathrm{ref}}(y' \mid x) \exp\big(r_i(y')/\beta_{\mathrm{eff}}\big)}.$$

If $\|r_1 - r_2\|_\infty \leq \delta$, then

$$D_{\mathrm{KL}}\big(\pi_1^\star \,\|\, \pi_2^\star\big) \leq \frac{2\delta}{\beta_{\mathrm{eff}}} \qquad \text{and} \qquad D_{\mathrm{KL}}\big(\pi_2^\star \,\|\, \pi_1^\star\big) \leq \frac{2\delta}{\beta_{\mathrm{eff}}}.$$

*Proof.* Fix $x$ and omit it from notation. Introduce the "potentials"

$$f(y) := \log \pi_{\mathrm{ref}}(y) + \frac{r_1(y)}{\beta_{\mathrm{eff}}}, \qquad g(y) := \log \pi_{\mathrm{ref}}(y) + \frac{r_2(y)}{\beta_{\mathrm{eff}}},$$

and the partition functions $Z_f = \sum_y \exp(f(y))$, $Z_g = \sum_y \exp(g(y))$. Then $\pi_1^\star(y) = \exp(f(y))/Z_f$ and $\pi_2^\star(y) = \exp(g(y))/Z_g$. We will bound $D_{\mathrm{KL}}(\pi_1^\star \| \pi_2^\star)$; the reverse bound follows by symmetry.

**Step 1: KL in terms of potentials.**   By definition,

$$D_{\mathrm{KL}}\big(\pi_1^\star\|\pi_2^\star\big) = \sum_y \pi_1^\star(y) \log \frac{\pi_1^\star(y)}{\pi_2^\star(y)} = \sum_y \pi_1^\star(y) \big(f(y) - g(y)\big) + \log Z_g - \log Z_f$$

$$= \mathbb{E}_{y \sim \pi_1^\star}\big[f(y) - g(y)\big] + \log Z_g - \log Z_f.$$

**Step 2: Bounding the expectation term.**   Since $\|r_1 - r_2\|_\infty \leq \delta$, we have

$$\|f - g\|_\infty = \left\|\tfrac{r_1 - r_2}{\beta_{\mathrm{eff}}}\right\|_\infty \leq \frac{\delta}{\beta_{\mathrm{eff}}}.$$

Thus

$$\mathbb{E}_{\pi_1^\star}[f - g] \leq \|f - g\|_\infty \leq \frac{\delta}{\beta_{\mathrm{eff}}}.$$

**Step 3: Lipschitzness of log-sum-exp.**   For any vectors $a, b$ on $\mathcal{Y}$,

$$\log \sum_y e^{b(y)} - \log \sum_y e^{a(y)} \leq \max_y \big(b(y) - a(y)\big),$$

which follows from $\sum_y e^{b(y)} = \sum_y e^{a(y)} e^{b(y) - a(y)} \leq e^{\max(b-a)} \sum_y e^{a(y)}$. By swapping $a, b$ one also has $\log \sum_y e^{a(y)} - \log \sum_y e^{b(y)} \leq \max_y \big(a(y) - b(y)\big)$. Hence

$$\big|\log Z_g - \log Z_f\big| \leq \|g - f\|_\infty \leq \frac{\delta}{\beta_{\mathrm{eff}}}.$$

**Step 4: Combine the bounds.**   Putting Steps 2–3 into Step 1 yields

$$D_{\mathrm{KL}}\big(\pi_1^\star\|\pi_2^\star\big) \leq \frac{\delta}{\beta_{\mathrm{eff}}} + \frac{\delta}{\beta_{\mathrm{eff}}} = \frac{2\delta}{\beta_{\mathrm{eff}}}.$$

Exchanging the roles of $(r_1, f, Z_f)$ and $(r_2, g, Z_g)$ gives $D_{\mathrm{KL}}(\pi_2^\star\|\pi_1^\star) \leq 2\delta/\beta_{\mathrm{eff}}$ as well.   $\square$

# E    ADDITIONAL DETAILS OF SFT TRAINING STAGE

## E.1    TRAINING DATASET DETAILS

To initialize the model with structured planning priors, we construct a supervised fine-tuning (SFT) dataset via response distillation from a high-capacity proprietary model. Specifically, we query *Gemini-2.0-flash* on a subset of EB-ALFRED tasks from the Embench benchmark and collect its response traces. For each task prompt, we record the full model output, including both intermediate reasoning and executable plan.

To ensure diversity and robustness, we repeat this process across three separate sampling rounds. The final dataset comprises a total of 4,245 training samples. Each sample includes a natural language instruction, egocentric image observations, and the model's full response in a structured format. An example training instance is shown in figure E.1

**SFT dataset example**

```
{
 "messages": [
  {
   "role": "user",
   "content": {EB-ALFRED prompt}
  },
  {
   "role": "assistant",
   "content": "{"reasoning_and_reflection": "The user wants me
   to rinse off a ladle and move it to the table. First I need
   to find the ladle, pick it up, find the sink, put the ladle
   down in the sink, find the faucet, turn on the faucet to
   rinse the ladle, turn off the faucet, pick up the ladle,
   find the table, and put the ladle on the table.",  "visual_
   state_description": "I can see a garbage can and a fridge in
    the current view.",  "language_plan": "First, locate a
   ladle and pick it up. Then, find a sink and put the ladle
   down in the sink. Next, find a faucet, turn it on to rinse
   the ladle, and then turn it off. After that, pick up the
   rinsed ladle, find a dining table, and place the ladle on
   the table.",  "executable_plan": [   {    "action_id": 64,
     "action_name": "find a Ladle"   },   {    "action_id":
   109,    "action_name": "pick up the Ladle"   },   {    "
   action_id": 79,    "action_name": "find a Sink"   },   {
   "action_id": 133,    "action_name": "put down the object in
   hand"   },   {    "action_id": 2,    "action_name": "find a
   Faucet"   },   {    "action_id": 155,    "action_name": "
   turn on the Faucet"   },   {    "action_id": 156,    "action
   _name": "turn off the Faucet"   },   {    "action_id": 64,
     "action_name": "find a Ladle"   },   {    "action_id":
   109,    "action_name": "pick up the Ladle"   },   {    "
   action_id": 18,    "action_name": "find a DiningTable"   },
     {    "action_id": 133,    "action_name": "put down the
   object in hand"   }]  }"
  }
 ],
 "images": [
  "example.png"
 ]
},
```

```
RFT dataset example

{
  {
    "id": "trial_T20190909_062150_965386_remain_0",
    "question": "{Our_RFT_prompt}",
    "answer": "['Goto handtowelholder', 'Pickup handtowel', '
    Goto garbagecan', 'Put handtowel']",
    "message": "[{\"role\": \"system\", \"content\": \"Solve the
     question. The user asks a question, and you solves it. You
    first thinks about the reasoning process in the mind and
    then provides the user with the answer.\"}, {\"role\": \"
    user\", \"content\": [{\"type\": \"image\", \"image\": \"
    example.jpg\"}, {\"type\": \"text\", \"text\": \"{Our_RFT_
    prompt}\"}]}]"
  }
},
```

## E.2 TRAINING HYPERPARAMETERS

We perform full-parameter supervised fine-tuning on the Qwen2.5-VL-7B model using the `LLaMA-Factory`(Zheng et al., 2024) framework. The training is conducted on 4 NVIDIA A100 40GB GPUs for approximately 8 hours. All hyperparameters are summarized in Table 3.

| Component | Setting | Component | Setting |
|---|---|---|---|
| *Model Configuration* | | | |
| image_max_pixels | 262144 | freeze_vision_tower | true |
| freeze_language_model | false | freeze_multi_modal_projector | true |
| deepspeed config | ds_z3_config.json | | |
| *Dataset Configuration* | | | |
| dataset | alfred_sft | template | qwen2_vl |
| cutoff_len | 2048 | max_samples | 1000 |
| overwrite_cache | true | preprocessing_workers | 16 |
| dataloader_workers | 4 | | |
| *Training Configuration* | | | |
| stage | sft | finetuning_type | full |
| do_train | true | num_train_epochs | 3.0 |
| learning_rate | 1e-5 | per_device_batch_size | 1 |
| grad_accum_steps | 2 | lr_scheduler | cosine |
| warmup_ratio | 0.1 | bf16 | true |
| ddp_timeout | 180000000 | | |

Table 3: Detailed hyperparameters used in supervised fine-tuning.

## E.3 TRAINING RESULTS

We record the final metrics and loss curve from the supervised fine-tuning process, as shown in Figure 5. The table summarizes key training statistics after 3 epochs of full-parameter tuning.

Figure 4: Summary of SFT training results.

| Metric | Value |
|--------|-------|
| Epochs | 3.0 |
| Total FLOPs | 3.13e13 |
| Training Loss | 0.252 |
| Runtime (s) | 21111.79 |
| Samples/sec | 0.142 |
| Steps/sec | 0.018 |

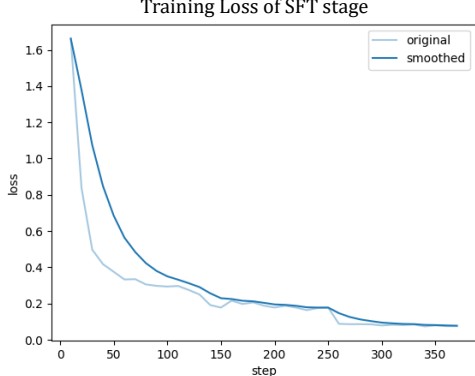

Figure 5: Training loss curve during SFT stage.

## F ADDITIONAL DETAILS OF RFT TRAINING STAGE

### F.1 TRAINING DATASET DETAILS

We construct our reinforcement fine-tuning (RFT) dataset based on the ALFRED benchmark, following the decomposition and formatting strategy described in Section 2. Notably, we do not reuse the SFT-distilled dataset for reinforcement fine-tuning. This decision is motivated by two key considerations: (1) the distilled data may contain suboptimal trajectories, introducing noise into the learning signal; (2) the distilled instruction format is tightly coupled with the benchmark evaluation prompts, whereas our constructed dataset introduces instruction variations that encourage greater policy generalization and better isolate the impact of reinforcement learning.

The resulting dataset contains 43,898 samples, each formatted to include a natural language instruction, a visual observation, and a ground-truth action sequence used for reward computation. We provide a full example of a training sample from the RFT dataset for reference in figureB.2

### F.2 FORMAT REWARD

To encourage valid and interpretable plans, we design a structured format reward inspired by Embench Yang et al. (2025), which requires the model's output to include four key sections: `reasoning_and_reflection`, `visual_state_description`, `language_plan`, and `executable_plan`. The reward is composed of three components:

$$R_{\text{format}} = R_{\text{structure}} + R_{\text{valid}} + R_{\text{match}}, \tag{16}$$

Each component reflects a specific aspect of format quality and all three components are weighted proportionally according to a 2:1:1 ratio:

- $R_{\text{structure}}$ rewards the presence of all required top-level fields, ensuring structural completeness.
- $R_{\text{valid}}$ measures the proportion of steps that include syntactically correct `action_id` and `action_name` pairs, reflecting output well-formedness.
- $R_{\text{match}}$ evaluates the number of actions that align with a predefined schema, ensuring semantic correctness and avoiding hallucinated actions.

### F.3 TRAINING HYPERPARAMETERS

We implement reinforcement fine-tuning using the `OpenRLHF`(Hu et al., 2024) framework, adopting the Generalized Reinforced Preference Optimization (GRPO) algorithm(Shao et al., 2024) to optimize policy learning from structured reward feedback. A full list of training hyperparameters is provided in Table 4.

| Hyperparameter | Value | Hyperparameter | Value |
|---|---|---|---|
| ref_num_nodes | 1 | vllm_num_engines | 8 |
| ref_num_gpus_per_node | 8 | actor_num_gpus_per_node | 8 |
| actor_num_nodes | 1 | vllm_tensor_parallel_size | 1 |
| vllm_gpu_memory_utilization | 0.65 | vllm_enable_sleep | True |
| vllm_sync_backend | nccl | temperature | 1.0 |
| max_epochs | 1 | max_episodes | 10 |
| prompt_max_len | 3000 | max_samples_len | 10000 |
| generate_max_len | 4096 | advantage_estimator | group_norm |
| zero_stage | 3 | actor_learning_rate | 1e-6 |
| init_kl_coef | 0.0 | n_samples_per_prompt | 8 |
| micro_train_batch_size | 1 | micro_rollout_batch_size | 2 |
| train_batch_size | 128 | rollout_batch_size | 128 |
| freeze_prefix | visual | enable_accuracy_filter | True |
| accuracy_lower_bound | 0.1 | accuracy_upper_bound | 0.9 |

Table 4: Hyperparameter configuration used during reinforcement fine-tuning.

## F.4 TRAINING LOG AND RESULT

We record the reinforcement fine-tuning process using several key indicators, as visualized in Figure 6.

The *total reward* refers to the combined score of the format reward and the accuracy reward. Due to the use of an online filtering strategy during training, we distinguish between two types of accuracy reward: *accuracy reward (filtered)*, which reflects the reward from selected high-quality samples that pass the filtering criteria, and *accuracy reward (original)*, which represents the average reward across all generated responses prior to filtering.

We also report two types of length statistics: *response length*, which quantifies the number of tokens generated by the model for each output, and *total length*, which denotes the combined token length of the input prompt and generated response.

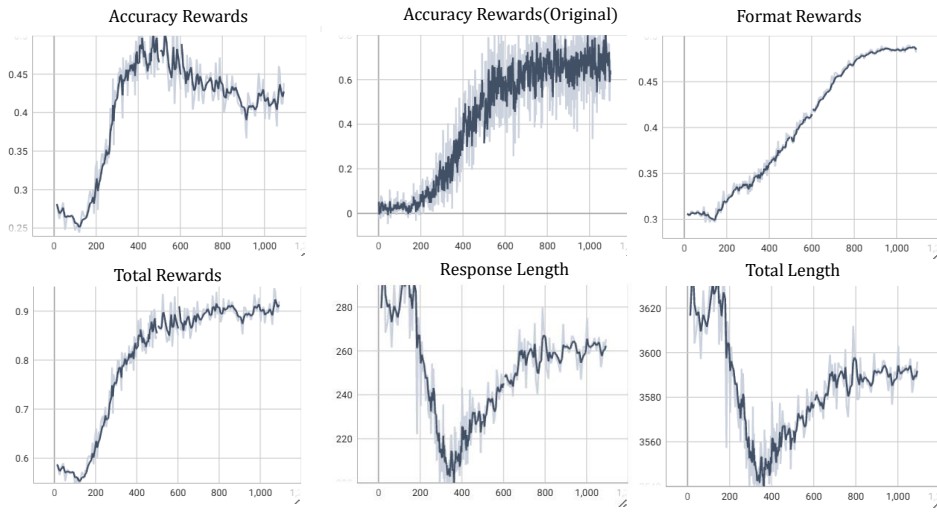

Figure 6: Training curve during reinforcement fine-tuning. The figure shows the progression of total reward, filtered and unfiltered accuracy reward, and generation length statistics.

# G  ADDITIONAL DETAILS FOR EVALUATION

## G.1  DETAILED INTRODUCTION TO EMBODIEDBENCH

**EmbodiedBench** is a comprehensive interactive benchmark designed to evaluate vision-language agents in embodied planning scenarios. Unlike static visual question answering settings, Embodied-Bench offers dynamic, simulation-based environments where agents must generate and execute multi-step plans grounded in first-person visual observations and natural language instructions. The benchmark spans four embodied environments and supports over 1,100 diverse tasks with hierarchical action levels, covering both high-level planning and low-level control.

In our work, we focus on two high-level planning environments within EmbodiedBench:

**EB-ALFRED.** EB-ALFRED is built upon the ALFRED dataset (Shridhar et al., 2020) and imple-mented on top of the AI2-THOR simulator (Kolve et al., 2017). It supports eight core skill types such as *pick up*, *put down*, *find*, *open/close*, and *turn on/off*. The environment provides egocentric visual inputs and textual feedback (e.g., success/failure messages), enabling agents to adaptively plan and act. Compared to the original ALFRED setup, EB-ALFRED enhances object diversity and simulator robustness. Specifically, it supports multiple object instances of the same type, merges redundant actions (e.g., unified *put down*), and dynamically adjusts the action space size (ranging from 171 to 298). These improvements provide a more realistic and flexible environment for assessing embodied planning capabilities.

**EB-Habitat.** EB-Habitat extends the Language Rearrangement benchmark (Savva et al., 2019), based on the Habitat 2.0 simulator. It focuses on five high-level skills: *navigation*, *pick*, *place*, *open*, and *close*. Unlike ALFRED, navigation in EB-Habitat is constrained to receptacle-type targets, requiring more sophisticated exploration and scene understanding. The environment includes 282 instruction templates and places more emphasis on spatial reasoning and location-aware planning, making it a complementary testbed for generalization.

**Task Subsets.** To enable fine-grained capability analysis, Embench introduces six distinct task subsets. Due to space limitations, we omit the subset "Long Horizon" from the main table and report its results in the Appendix.

- **Base:** Evaluates standard task-solving skills under low to medium complexity, testing general planning competence.
- **Common Sense:** Assesses agents' ability to reason over implicit object references and everyday knowledge.
- **Complex Instruction:** Presents long, noisy or ambiguous contexts to evaluate the agent's ability to extract user intent.
- **Spatial Awareness:** Requires understanding object relationships in space, such as relative positions or arrangements.
- **Visual Appearance:** Involves identifying objects via attributes like color or shape, testing fine-grained visual recognition.
- **Long Horizon:** Contains tasks demanding long sequences of actions (often exceeding 15 steps), stressing planning depth and temporal consistency.

Each subset is designed to probe a specific capability of embodied reasoning, such as commonsense inference, spatial understanding, or long-horizon planning. In our experiments, we evaluate model performance across all six subsets to provide a fine-grained analysis. As shown in Table 5, these categories span a wide range of reasoning challenges. Notably, since our reinforcement fine-tuning dataset only includes *Base* tasks, we observe a significantly larger performance gain in this category, whereas improvements in other subsets are relatively modest. This highlights the need for more diverse training data to support generalizable planning across varied task types.

Overall, Embench provides a rigorous, scalable, and diagnostic framework for benchmarking embod-ied agents across diverse real-world challenges. In our setup, we use EB-ALFRED for in-domain training and evaluation, while EB-Habitat serves as an out-of-domain testbed to examine generaliza-tion performance.

Table 5: Examples of each task type from EB-ALFRED and EB-Habitat.

| Task Subset | ALFRED Example | Habitat Example |
|---|---|---|
| Base | Put washed lettuce in the refrigerator. | Move one of the pear items to the indicated sofa. |
| Common Sense | Place washed leafy green vegetable in a receptacle that can keep it fresh. | Prepare for a game by delivering something to play with to the TV stand. |
| Complex Instruction | Place the washed lettuce in the refrigerator. This way, it's ready for any delightful recipe ideas you have. | When you find the fridge door open, go ahead and move one bowl to the sofa; otherwise, transport one hammer to the sofa. |
| Spatial Awareness | Put two spray bottles in the cabinet under the sink against the wall. | Move a spatula from the right counter to the right receptacle of the left counter. |
| Visual Appearance | Put a knife in a blue container onto the black table in the corner. | Deliver a small red object with green top to the indicated large gray piece of furniture. |
| Long Horizon | Pick up knife, slice apple, put knife in bowl, heat apple slice in microwave, put apple slice on table. | Move the rubrics cube to the left counter; the towel to the left counter, and the bowl to the brown table. |

## G.2 DETAILED INTRODUCTION TO BASELINES

To comprehensively evaluate our proposed method, we compare it against a diverse set of baselines, covering both proprietary and open-source models, as well as models specifically optimized for multimodal reasoning and embodied planning.

(1) *Closed-source models*: we include several leading proprietary vision-language models as strong general-purpose baselines, including Claude-3.5-Sonnet(cla), Gemini-2.0-flash(gem), GPT-4o(gpt, a), and GPT-4o-mini(gpt, b).

(2) *Open-source general VLMs*: we evaluate widely adopted open-source VLMs trained for generic multimodal tasks, such as LLaMA-3.2-Vision-11B(lla), Qwen2.5-VL-7B(Bai et al., 2025) and InternVL2.5-8B(Chen et al., 2024).

(3) *Open-source reasoning VLMs*: we further include two representative models that have been explicitly optimized for multimodal reasoning, including MM-Eureka(Meng et al., 2025) and R1-VL(Zhang et al., 2025a).

MM-Eureka extends rule-based reinforcement learning to multimodal reasoning, enabling models to improve through reward-driven optimization without supervised fine-tuning. It reproduces key behaviors from language-only RL systems, such as reflection and reward-aligned response growth, achieving strong data efficiency and reasoning performance.

R1-VL enhances step-by-step reasoning in multimodal LLMs via StepGRPO, a reinforcement learning framework with dense, rule-based rewards for accuracy and logical consistency. It surpasses imitation learning by guiding models to self-correct flawed reasoning, achieving superior results on multiple benchmarks.

We also attempted to evaluate other open-source reasoning models, such as VisualRFT(Liu et al., 2025) and Open-R1(Face, 2025). However, their inference speed was prohibitively slow, resulting in impractically long evaluation time on interactive benchmarks. Additionally, their final planning performance remained poor for embodied planning scenarios.

(4) *Embodied VLMs*: we also include RoboBrain(Ji et al.) and TAPA(Wu et al., 2023), two representative open-source large models designed for embodied tasks.

TAPA is the first model specifically optimized for embodied multi-step planning, but it lacks visual perception capability; thus, we convert visual observations into textual descriptions for evaluation.

RoboBrain is a state-of-the-art VLM for embodied scenarios that integrates robotic and general multimodal data through a multi-stage training pipeline,leveraging long-horizon video and high-resolution image supervision to enhance manipulation and planning performance.

| Model | EB-ALFRED (Seen) | | | | | | | | | | | | | |
|---|---|---|---|---|---|---|---|---|---|---|---|---|---|---|
| | Avg | | Base | | Common | | Complex | | Visual | | Spatial | | Long | |
| | PR | ES | PR | ES | PR | ES | PR | ES | PR | ES | PR | ES | PR | ES |
| **Closed-Source MLLMs** | | | | | | | | | | | | | | |
| Claude-3.5-Sonnet | 70.11 | 14.9 | 72.67 | 12.2 | 65.83 | 12.74 | 73.33 | 11.48 | 65.5 | 14.02 | 68.83 | 16.96 | 74.5 | 21.98 |
| Gemini-2.0-flash | 57.13 | 16.5 | 61.83 | 13.96 | 60.67 | 14.0 | 55.33 | 15.16 | 55.33 | 15.26 | 46.67 | 17.04 | 63.0 | 23.56 |
| GPT-4o | 61.78 | 16.77 | 65.67 | 12.54 | 57.17 | 16.1 | 74.67 | 13.92 | 58.33 | 15.2 | 52.33 | 17.58 | 62.5 | 25.43 |
| GPT-4o-mini | 30.42 | 19.69 | 36.33 | 17.32 | 29.83 | 18.06 | 38.0 | 17.74 | 27.33 | 18.48 | 31.0 | 19.9 | 20.0 | 26.62 |
| **Open-Source General MLLMs** | | | | | | | | | | | | | | |
| Qwen2.5-VL-7B | 6.86 | 9.4 | 5.67 | 8.78 | 4.0 | 4.2 | 5.0 | 5.28 | 5.33 | 7.16 | 0.67 | 8.26 | 20.5 | 22.72 |
| InternVL2.5-8B | 5.78 | 7.87 | 6.17 | 8.2 | 0.67 | 4.9 | 16.0 | 8.92 | 4.0 | 6.78 | 6.33 | 7.52 | 1.5 | 10.92 |
| **Open-Source Reasoning MLLMs** | | | | | | | | | | | | | | |
| R1-VL-7B | 2.78 | 4.01 | 3.0 | 3.22 | 3.0 | 2.06 | 6.0 | 1.7 | 0.67 | 1.62 | 0.0 | 2.66 | 4.0 | 12.78 |
| MM-Eureka-Qwen-7B | 6.59 | 8.48 | 8.67 | 7.64 | 5.33 | 5.04 | 8.67 | 9.72 | 3.67 | 6.46 | 0.67 | 6.58 | 12.5 | 15.42 |
| **Open-Source Embodied MLLMs** | | | | | | | | | | | | | | |
| RoboBrain | 1.22 | 6.7 | 3.33 | 6.1 | 0.67 | 6.3 | 0.67 | 3.68 | 0.67 | 7.56 | 0 | 6.36 | 2.0 | 10.22 |
| Tapa | 0 | 0.03 | 0 | 0.06 | 0 | 0 | 0 | 0 | 0 | 0.04 | 0 | 0.08 | 0 | 0 |
| **Open-Source Embodied + Reasoning MLLMs** | | | | | | | | | | | | | | |
| Ours (Base) | 6.86 | 9.4 | 5.67 | 8.78 | 4.0 | 4.2 | 5.0 | 5.28 | 5.33 | 7.16 | 0.67 | 8.26 | 20.5 | 22.72 |
| Ours (SFT only) | 23.8 | 15.06 | 39 | 13.14 | 26.6 | 13.04 | 27.6 | 12.56 | 19.3 | 14.12 | 14.3 | 15.16 | 16.5 | 22.38 |
| Ours (SFT+RFT) | 53.03 | 17.63 | 70.3 | 13.72 | 65 | 15.3 | 59.9 | 16.12 | 48.5 | 17.06 | 43 | 16.88 | 31.5 | 26.7 |

Table 6: Progress Rate (PR) and Environment Steps (ES) on EB-ALFRED (Seen)

While there exist other VLMs designed for embodied settings, many of them are unavailable for public use, such as ReasonRFT(Tan et al., 2025),Shi et al. (2025a) and Embodied-Reasoner(Zhang et al., 2025b). Other models, such as EmbodiedGPT(Mu et al.) and TAPA(Wu et al., 2023), exhibit poor generalization to new task distributions, achieving near-zero scores on Embench tasks and revealing a lack of transferable planning capabilities.

### G.3 EXPERIMENT RESULTS USING SUPPLEMENTARY METRICS

In addition to task success rate, we provide supplementary evaluation results using two additional metrics: **Progress Rate (PR)** and **Environment Steps (ES)**.

**Progress Rate (PR)** quantifies the degree to which the agent completes the task, measured as the proportion of goal conditions satisfied by the final environment state. This metric provides a finer-grained signal than binary success, especially for partially completed tasks.

**Environment Steps (ES)** refers to the number of actions executed in the environment before task termination. A lower ES generally indicates more efficient planning and fewer redundant or failed actions.

Complete results across these metrics are reported in Appendix Tables 6 and 7.

## H LIMITATION AND FUTURE WORK

A natural next step is to extend our offline reward paradigm beyond embodied planning, applying it to broader scenarios such as vision-language-action (VLA) models and other multi-turn reasoning tasks. In these settings, a hybrid approach that combines offline trajectory-based rewards with selective online feedback may further enhance adaptability while retaining stability.

In addition, our current focus lies on high-level embodied planning in simulation, producing structured action sequences that guide downstream controllers. While our method already demonstrates strong performance and generalization in benchmarks, deploying it on physical robotic platforms remains an important avenue for validation and integration with low-level control systems.

| Model | EB-Habitat (Unseen) | | | | | | | | | | | | | |
| | Avg | | Base | | Common | | Complex | | Visual | | Spatial | | Long | |
| | PR | ES | PR | ES | PR | ES | PR | ES | PR | ES | PR | ES | PR | ES |
| --- | --- | --- | --- | --- | --- | --- | --- | --- | --- | --- | --- | --- | --- | --- |
| **Closed-Source MLLMs** | | | | | | | | | | | | | | |
| Claude-3.5-Sonnet | 70.9 | 10.7 | 98 | 6.54 | 69.5 | 10.46 | 75.5 | 10.6 | 75.1 | 10.74 | 45.2 | 9.44 | 62.1 | 16.42 |
| Gemini-2.0-flash | 38.5 | 13.41 | 76.5 | 8.56 | 31.5 | 12.9 | 34 | 15.66 | 32.7 | 13.7 | 37 | 12 | 19.8 | 17.66 |
| GPT-4o | 60.8 | 14.32 | 85.3 | 9.76 | 34 | 14.74 | 67.5 | 13.34 | 64.3 | 13.82 | 46.3 | 14.78 | 67.2 | 19.5 |
| GPT-4o-mini | 44.2 | 18.8 | 73.6 | 10.96 | 46 | 18.78 | 40.5 | 19.76 | 36.8 | 21.76 | 47.5 | 18.86 | 20.6 | 22.7 |
| **Open-Source General MLLMs** | | | | | | | | | | | | | | |
| Qwen2.5-VL-7B | 19.05 | 12.58 | 44.5 | 10.64 | 6.5 | 14.9 | 17 | 11.12 | 6.4 | 14.12 | 28.8 | 11.74 | 11.1 | 12.94 |
| InternVL2.5-8B | 26 | 16.77 | 52.9 | 13.1 | 13 | 19.1 | 22 | 16.48 | 21.6 | 18.36 | 35.4 | 18.24 | 11.1 | 15.32 |
| **Open-Source Reasoning MLLMs** | | | | | | | | | | | | | | |
| R1-VL-7B | 8.06 | 5.08 | 24.6 | 5.9 | 0 | 3.78 | 4 | 4.38 | 6 | 1.8 | 11.8 | 7.78 | 2 | 6.88 |
| MM-Eureka-Qwen-7B | 22.03 | 13.53 | 40.5 | 10.24 | 20.5 | 15.78 | 19 | 11.34 | 15.9 | 15.66 | 31.3 | 13.74 | 5 | 14.4 |
| **Open-Source Embodied MLLMs** | | | | | | | | | | | | | | |
| RoboBrain | 20.18 | 10.68 | 39.1 | 8.08 | 9.5 | 9.08 | 21 | 11.3 | 12.9 | 13.9 | 31.1 | 11.48 | 7.5 | 10.24 |
| Tapa | 0 | 0 | 0 | 0 | 0 | 0 | 0 | 0 | 0 | 0 | 0 | 0 | 0 | 0 |
| **Open-Source Embodied + Reasoning MLLMs** | | | | | | | | | | | | | | |
| Ours (Base) | 19.05 | 12.58 | 44.5 | 10.64 | 6.5 | 14.9 | 17 | 11.12 | 6.4 | 14.12 | 28.8 | 11.74 | 11.1 | 12.94 |
| Ours (SFT only) | 20.05 | 12.40 | 38.75 | 10.62 | 7 | 12.3 | 19.5 | 12.76 | 16 | 11.24 | 34.6 | 15.26 | 4.5 | 12.26 |
| Ours (SFT+RFT) | 27.18 | 13.31 | 58.75 | 8.72 | 15 | 14.98 | 23 | 13.3 | 20 | 13.36 | 37 | 13.78 | 9.33 | 15.76 |

Table 7: Progress Rate (PR) and Environment Steps (ES) on EB-Habitat (Unseen)

# I CASE STUDY AND VISUALIZATION

## I.1 CASE STUDY

To better understand how our model performs embodied multi-step planning, we present detailed case studies illustrating its behavior and reasoning process. Specifically, we compare the outputs of our reinforcement-tuned model with the base Qwen2.5-VL model to highlight improvements in planning coherence and action correctness, we also present full multi-step execution trajectories from our model to show how it plans and interacts with the environment to complete specific tasks.

Figure 7 and Figure 8 show side-by-side comparisons between the two models in the EB-ALFRED and EB-Habitat environments, respectively. We observe that the base model often produces incomplete or illogical plans, while our model generates more structured and context-aware action sequences, along with interpretable reasoning steps.

Figure 9, Figure 10,Figure 11 and Figure 12 further visualize full planning trajectories executed by our model in representative tasks from EB-ALFRED and EB-Habitat. These examples demonstrate the model's ability to maintain long-horizon coherence, correctly interpret dynamic observations, and recover from intermediate failures.

## I.2 PROMPT

In this section, we document the full prompt formats used in both evaluation and training stages, including for EB-ALFRED, EB-Habitat, and our reinforcement fine-tuning (RFT) process.

**EB-ALFRED Prompt.** The EB-ALFRED prompt is used for evaluating models within the EB-ALFRED environment of Embench. Our SFT stage also adopts this prompt format.

**EB-Habitat Prompt.** This prompt format is used in Embench's EB-Habitat environment, which differs from EB-ALFRED in simulator, object distribution, and language patterns.

**RFT Training Prompt.** During reinforcement fine-tuning, we adopt a custom prompt format. While still grounded in the same simulation environment, our RFT prompts include modifications in

action representation and instruction phrasing. These differences help introduce broader data diversity and encourage the model to learn a more generalizable planning policy.

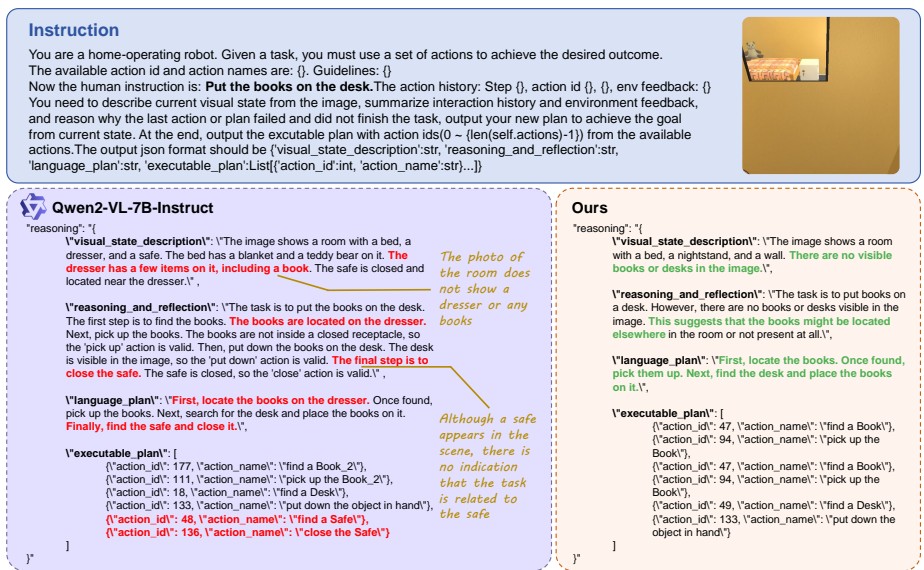

Figure 7: Case comparison (Our model vs Base model) in EB-ALFRED

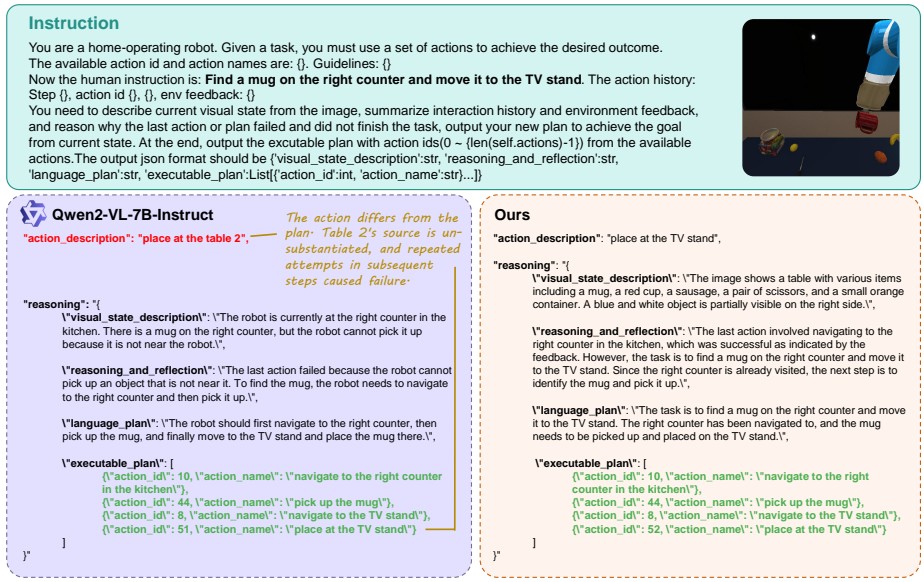

Figure 8: Case comparison (Our model vs Base model) in EB-Habitat

| Instruction | Examine a tv remote next to the light of a tall lamp. |
|---|---|

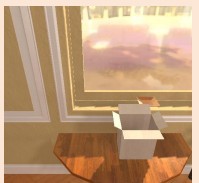

"reasoning": "{"visual_state_description": "The image shows a room with a wooden table, a box on top of it, and a window with sunlight coming through. There is a tall lamp near the window, but no TV remote is visible in the scene.", "reasoning_and_reflection": "The task requires examining a TV remote under the light of a tall lamp. However, there is no TV remote visible in the image. The lamp is present, but the TV remote is not. This suggests that the TV remote might be out of view or not in the current scene.", "language_plan": "First, locate the TV remote. Once found, pick it up. Next, find the tall lamp and turn it on to examine the remote.", "executable_plan": [{"action_id": 68, "action_name": "find a RemoteControl"}, {"action_id": 116, "action_name": "pick up the RemoteControl"}, {"action_id": 27, "action_name": "find a FloorLamp"}, {"action_id": 153, "action_name": "turn on the FloorLamp"}]}"

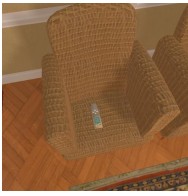

### Action 1

"env_step": 1
"env_feedback": "Last action executed successfully."
"action_id": 68
"action_description": "find a RemoteControl"

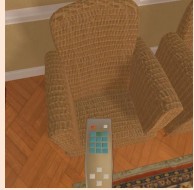

### Action 2

"env_step": 2
"env_feedback": "Last action executed successfully."
"action_id": 116
"action_description": "pick up the RemoteControl"

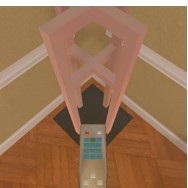

### Action 3

"env_step": 3
"env_feedback": "Last action executed successfully."
"action_id": 27
"action_description": "find a FloorLamp"

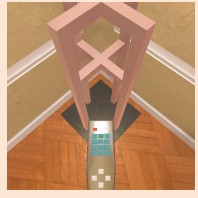

### Action 4

"env_step": 4
"env_feedback": "Last action executed successfully."
"action_id": 153
"action_description": "turn on the FloorLamp"

| | Task Success |
|---|---|

Figure 9: Our model's Full Trajectory execution in EB-ALFRED

| Instruction | Retrieve a apple and place it into the predetermined sofa. |
|---|---|

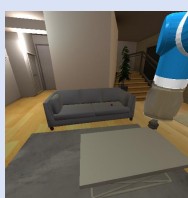

"reasoning": "{"visual_state_description": "The image shows a living room with a gray sofa, a coffee table, a staircase, and a blue object resembling a person. The apple is not visible in the image.", "reasoning_and_reflection": "The task is to retrieve an apple and place it on the sofa. However, the apple is not visible in the image. This suggests that the apple might be located elsewhere in the house, possibly in the kitchen or another room. The current plan should involve navigating to the kitchen to find the apple and then placing it on the sofa.", "language_plan": "The task is to find an apple and place it on the sofa. Since the apple is not visible in the current room, the robot should navigate to the kitchen to search for the apple.", "executable_plan": [{"action_id": 10, "action_name": "navigate to the right counter in the kitchen"}, {"action_id": 33, "action_name": "pick up the apple"}, {"action_id": 12, "action_name": "navigate to the sofa"}, {"action_id": 56, "action_name": "place at the sofa"}]}"

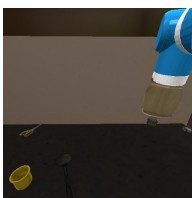

**Action 1**

"env_step": 1
"env_feedback": "Last action executed successfully."
"action_id": 10
"action_description": "navigate to the right counter in the kitchen"

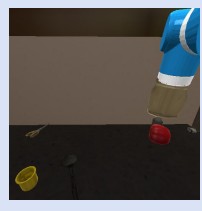

**Action 2**

"env_step": 2
"env_feedback": "Last action executed successfully and you are holding apple."
"action_id": 33
"action_description":  "pick up the apple"

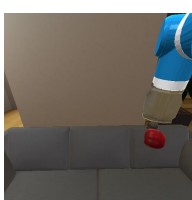

**Action 3**

"env_step": 3
"env_feedback": "Last action executed successfully."
"action_id": 12
"action_description": "navigate to the sofa"

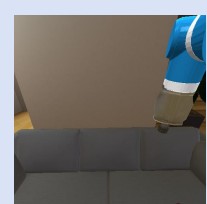

**Action 4**

"env_step": 4
"env_feedback": "Last action executed successfully and you are holding nothing."
"action_id": 56
"action_description": "place at the sofa"

| | *Task* *Success* |
|---|---|

Figure 10: Our model's Full Trajectory execution example 1 in EB-Habitat

| **Instruction** | Move a ball to the sofa. |
|---|---|

"reasoning": "{"visual_state_description": "The image shows a living room with a sofa, a TV stand, and a table. A ball is located on the floor near the TV stand.", "reasoning_and_reflection": "The task is to move a ball from the floor to the sofa. The ball is clearly visible on the floor near the TV stand. The next step is to pick up the ball and then place it on the sofa.", "language_plan": "The plan is to first navigate to the ball, pick it up, and then place it on the sofa.", "executable_plan": [{"action_id": 8, "action_name": "navigate to the ball"}, {"action_id": 16, "action_name": "pick up the ball"}, {"action_id": 12, "action_name": "navigate to the sofa"}, {"action_id": 56, "action_name": "place at the sofa"}]}"

### Action 1

"env_step": 1
"env_feedback": "Last action executed successfully."
"action_id": 8
"action_description": "navigate to the TV stand"

### Action 2

"env_step": 2
"env_feedback": "Last action is invalid. Robot cannot pick any object that is not near the robot. Navigate to other place to find the object."
"action_id": 16
"action_description": "pick up the ball"

*Replanning*

"reasoning": "{"visual_state_description": "The image shows a room with a plant in a pot on the floor, a sofa, and a table with some items on it. The ball is located on the floor next to the plant.", "reasoning_and_reflection": "The previous action was to navigate to the TV stand, but the ball was not found there. The ball is now visible on the floor next to the plant. The robot needs to pick up the ball and move it to the sofa.", "language_plan": "The robot should navigate to the ball, pick it up, and then move to the sofa to place the ball there.", "executable_plan": [{"action_id": 8, "action_name": "navigate to the TV stand"}, {"action_id": 16, "action_name": "pick up the ball"}, {"action_id": 12, "action_name": "navigate to the sofa"}, {"action_id": 56, "action_name": "place at the sofa"}]}"

### Action 3

"env_step": 3
"env_feedback": "Last action executed successfully."
"action_id": 8
"action_description": "navigate to the TV stand"

### Action 4

"env_step": 4
"env_feedback": "Last action is invalid. Robot cannot pick any object that is not near the robot. Navigate to other place to find the object."
"action_id": 16
"action_description": "pick up the ball"

*Replanning*

"reasoning": "{"visual_state_description": "The image shows a room with a plant in a pot, a blue and white object, a book, and a red and green object on the floor. The ball is not visible in the image.", "reasoning_and_reflection": "The previous attempts to pick up the ball were unsuccessful due to the robot being too far away from the ball. The ball is likely located on the floor, but the robot has not yet navigated to the correct position to pick it up. The current plan should involve navigating to the floor to find and pick up the ball.", "language_plan": "The robot needs to navigate to the floor to find the ball and then pick it up.", "executable_plan": [{"action_id": 6, "action_name": "navigate to the floor"}, {"action_id": 16, "action_name": "pick up the ball"}]}"

Figure 11: Our model's Full Trajectory execution example 2 (part 1) in EB-Habitat

**Action 5**

"env_step": 5
"env_feedback": "Last action executed successfully."
"action_id": 6
"action_description": "navigate to the table 1"

**Action 6**

"env_step": 6
"env_feedback": "Last action executed successfully and you are holding ball."
"action_id": 16
"action_description": "pick up the ball"

*Replanning*

"reasoning": "{"visual_state_description": "The robot is currently standing in front of the table 1, holding a ball. The ball is located on the table 1, along with a toy airplane, a cup, a bowl, and a lemon. The sofa is visible in the background.", "reasoning_and_reflection": "The robot has already picked up the ball from the table 1. The next step is to navigate to the sofa and place the ball there. The previous attempts to pick up the ball were successful, but the robot needs to move to the sofa to complete the task.", "language_plan": "The robot needs to navigate to the sofa and place the ball there.", "executable_plan": [{"action_id": 12, "action_name": "navigate to the sofa"}, {"action_id": 56, "action_name": "place at the sofa"}]}"

**Action 7**

"env_step": 7
"env_feedback": "Last action executed successfully."
"action_id": 12
"action_description": "navigate to the sofa"

**Action 8**

"env_step": 8
"env_feedback": "Last action executed successfully and you are holding nothing."
"action_id": 56
"action_description": "place at the sofa"

*Task Success*

Figure 12: Our model's Full Trajectory execution example 2 (part 2) in EB-Habitat

## EB-ALFRED prompt

"<image>## You are a robot operating in a home. Given a task, you must accomplish the task using a defined set of actions to achieve the desired outcome.

## Action Descriptions and Validity Rules* Find: Parameterized by the name of the receptacle to navigate to. So long as the object is present in the scene, this skill is always valid * Pick up: Parameterized by the name of the object to pick. Only valid if the robot is close to the object, not holding another object, and the object is not inside a closed receptacle.* Put down: Parameterized by the name of the object to put down to a nearby receptacle. Only valid if the robot is holding an object. * Drop: Parameterized by the name of the object to put down. It is different from Put down action, as this does not guarantee the held object will be put into a specified receptacle. * Open: Parameterized by the name of the receptacle to open. Only valid if the receptacle is closed and the robot is close to the receptacle. * Close: Parameterized by the name of the receptacle to close. Only valid if the receptacle is open and the robot is close to the receptacle. * Turn on: Parameterized by the name of the object to turn on. Only valid if the object is turned off and the robot is close to the object. * Turn off: Parameterized by the name of the object to turn off. Only valid if the object is turned on and the robot is close to the object. * Slice: Parameterized by the name of the object to slice. Only valid if the object is sliceable and the robot is close to the object.

## The available action id (0 ~ 207) and action names are: { ALFRED ACTION LIST}

## Task Execution Example:{IN-CONTEXT TASK EXAMPLE}

## Guidelines 1. **Output Plan**: Avoid generating empty plan. Each plan should include no more than 20 actions. 2. **Visibility**: Always locate a visible object by the 'find' action before interacting with it. 3. **Action Guidelines**: Make sure match the action name and its corresponding action id in the output.newline Avoid performing actions that do not meet the defined validity criteria. For instance, if you want to put object in a receptacle, use 'put down' rather than 'drop' actions. 4. **Prevent Repeating Action Sequences**: Do not repeatedly execute the same action or sequence of actions. Try to modify the action sequence because previous actions do not lead to success. 5. **Multiple Instances**: There may be multiple instances of the same object, distinguished by an index following their names, e.g., Cabinet_2, Cabinet_3. You can explore these instances if you do not find the desired object in the current receptacle. 6. **Reflection on History and Feedback**: Use interaction history and feedback from the environment to refine and improve your current plan. If the last action is invalid, reflect on the reason, such as not adhering to action rules or missing preliminary actions, and adjust your plan accordingly.

## Now the human instruction is: Rinse off a ladle and move it to the table. You are supposed to output in json. You need to describe current visual state from the image, output your reasoning steps and plan. At the end, output the action id (0 ~ 207) from the available actions to excute."

1566
1567
1568
1569
1570
1571
1572
1573
1574
1575
1576
1577
1578
1579
1580
1581
1582
1583
1584
1585
1586
1587
1588
1589
1590
1591
1592
1593
1594
1595
1596
1597
1598
1599
1600
1601
1602
1603
1604
1605
1606
1607
1608
1609
1610
1611
1612
1613
1614
1615
1616
1617
1618
1619

---

**EB-Habitat prompt**

```
<image>##You are a robot operating in a home. Given a task,
  you must accomplish the task using a defined set of actions
  to achieve the desired outcome.

## Action Descriptions and Validity Rules: * Navigation:
  Parameterized by the name of the receptacle to navigate to.
  So long as the receptacle is present in the scene, this
  skill is always valid. * Pick: Parameterized by the name of
  the object to pick. Only valid if the robot is close to the
  object, not holding another object, and the object is not
  inside a closed receptacle. * Place: Parameterized by the
  name of the receptacle to place the object on. Only valid if
   the robot is close to the receptacle and is holding an
  object. * Open: Parameterized by the name of the receptacle
  to open. Only valid if the receptacle is closed and the
  robot is close to the receptacle. * Close: Parameterized by
  the name of the receptacle to close. Only valid if the
  receptacle is open and the robot is close to the receptacle.

## The available action id (0 ~ 69) and action names are:{
  HABITAT ACTION LIST}

## Task Execution Example:{IN-CONTEXT TASK EXAMPLE}

## Guidelines 1. **Output Plan**: Avoid generating empty plan.
   Each plan should include no more than 20 actions. 2. **
  Visibility**: If an object is not currently visible, use the
   \"Navigation\" action to locate it or its receptacle before
   attempting other operations. 3. **Action Validity**: Make
  sure match the action name and its corresponding action id
  in the output. Avoid performing actions that do not meet the
   defined validity criteria. 4. **Prevent Repeating Action
  Sequences**: Do not repeatedly execute the same action or
  sequence of actions. Try to modify the action sequence
  because previous actions do not lead to success. 5. **
  Multiple Instances**: There may be multiple instances of the
   same object, distinguished by an index following their
  names, e.g., cabinet 2, cabinet 3. You can explore these
  instances if you do not find the desired object in the
  current receptacle. 6. **Reflection on History and Feedback
  **: Use interaction history and feedback from the
  environment to refine and enhance your current strategies
  and actions. If the last action is invalid, reflect on the
  reason, such as not adhering to action rules or missing
  preliminary actions, and adjust your plan accordingly.

## Now the human instruction is: Move one of the pear items to
   the indicated sofa. You are supposed to output in json. You
   need to describe current visual state from the image,
  output your reasoning steps and plan. At the end, output the
   action id (0 ~ 69) from the available actions to excute."
```

**Our RFT prompt**

You are a robot operating in a home. Given a task, you must accomplish the task using a defined set of actions to achieve the desired outcome.

## Action Descriptions and Validity Rules * GotoLocation: Parameterized by the name of the target location or receptacle to navigate to. Always valid so long as the target exists in the scene. * PickupObject: Parameterized by the name of the object to pick up. Valid only if the robot is close to the object, is not holding anything, and the object is accessible. * PutObject: Parameterized by the name of the receptacle or surface where the held object will be placed. Valid only if the robot is holding an object. * ToggleObject: Parameterized by the name of the object whose state can be toggled (e.g., lamp, faucet). Valid only if the robot is close to the object. * CoolObject: Parameterized by the name of the object to cool. Requires the robot to be holding the object and near a cooling appliance such as a fridge. * SliceObject: Parameterized by the name of the object to slice. Requires that the object is slice-able and the robot holds an appropriate cutting tool. * CleanObject: Parameterized by the name of the object to clean. Requires the robot to be near a water source and the object supports cleaning. * HeatObject: Parameterized by the name of the object to heat. Requires the robot to be holding the object and near a heating appliance such as a microwave or stove.

## The available action id (0 ~ 224) and action names are:{OUR RFT ACTION LIST}

## Guidelines 1. **Output Plan**: Avoid generating empty plan. Each plan should include no more than 20 actions. 2. **Visibility**: Always locate a visible object by the 'goto' action before interacting with it. 3. **Action Guidelines**: Make sure the action name and its corresponding action id match in the output. Avoid performing actions that do not meet the defined validity criteria. 4. **Prevent Repeating Action Sequences**: Do not repeatedly execute the same action or sequence of actions. 5. **Multiple Instances**: There may be multiple instances of the same object, distinguished by an index following their names, e.g., Cabinet_2. 6. **Reflection on History and Feedback**: Use interaction history and feedback from the environment to refine and improve your current plan.

## Expected JSON output format```json {\"reasoning_and_reflection\": \"<string>\", \"visual_state_description\": \"<string>\", \"language_plan\": \"<string>\", \"executable_plan\": [ {\"action_id\": <int>, \"action_name\": \"<string>\"} ]}```

## Now the human instruction is: put a towel into a garbage can The history actions are: [{HISTORY LIST}] newlineConsidering the above interaction history and the current image state, to achieve the human instruction.newlineYou are supposed to output in json. You need to describe current visual state from the image, output your reasoning steps and plan. You shuold think carefully and output the comprehensive thought process in 'reasoning_and_reflection' part. At the end, output the action id (0 ~ 224) from the available actions to execute."

1674
1675
1676
1677
1678
1679
1680
1681
1682
1683
1684
1685
1686
1687
1688
1689
1690
1691
1692
1693
1694
1695
1696
1697
1698
1699
1700
1701
1702
1703
1704
1705
1706
1707
1708
1709
1710
1711
1712
1713
1714
1715
1716
1717
1718
1719
1720
1721
1722
1723
1724
1725
1726
1727

## Part of EB-ALFRED Action list

action id 1: find a Potato, action id 2: find a Faucet, action
    id 3: find a Ottoman, action id 4: find a CoffeeMachine,
    action id 5: find a Candle, action id 6: find a CD, action
    id 7: find a Pan, action id 8: find a Watch, action id 9:
    find a HandTowel, action id 10: find a SprayBottle, action
    id 11: find a BaseballBat, action id 12: find a CellPhone,
    action id 13: find a Kettle, action id 14: find a Mug,
    action id 15: find a StoveBurner, action id 16: find a Bowl,
     action id 17: find a Toilet, action id 18: find a
    DiningTable, action id 19: find a Spoon, action id 20: find
    a TissueBox, action id 21: find a Shelf, action id 22: find
    a Apple, action id 23: find a TennisRacket, action id 24:
    find a SoapBar, action id 25: find a Cloth, action id 26:
    find a Plunger, action id 27: find a FloorLamp, action id
    28: find a ToiletPaperHanger, action id 29: find a
    CoffeeTable, action id 30: find a Spatula, action id 31:
    find a Plate, action id 32: find a Bed, action id 33: find a
     Glassbottle, action id 34: find a Knife, action id 35: find
     a Tomato, action id 36: find a ButterKnife, action id 37:
    find a Dresser, action id 38: find a Microwave, action id
    39: find a CounterTop, action id 40: find a GarbageCan,
    action id 41: find a WateringCan, action id 42: find a Vase,
     action id 43: find a ArmChair, action id 44: find a Safe,
    action id 45: find a KeyChain, action id 46: find a Pot,
    action id 47: find a Pen, action id 48: find a Cabinet,
    action id 49: find a Desk, action id 50: find a Newspaper,
    action id 51: find a Drawer, action id 52: find a Sofa,
    action id 53: find a Bread, action id 54: find a Book,
    action id 55: find a Lettuce, action id 56: find a
    CreditCard, action id 57: find a AlarmClock, action id 58:
    find a ToiletPaper, action id 59: find a SideTable, action
    id 60: find a Fork, action id 61: find a Box, action id 62:
    find a Egg, action id 63: find a DeskLamp, action id 64:
    find a Ladle, action id 65: find a WineBottle, action id 66:
     find a Pencil, action id 67: find a Laptop, action id 68:
    find a RemoteControl, action id 69: find a BasketBall,
    action id 70: find a DishSponge, action id 71: find a Cup,
    action id 72: find a SaltShaker, action id 73: find a
    PepperShaker, action id 74: find a Pillow, action id 75:
    find a Bathtub, action id 76: find a SoapBottle, action id
    77: find a Statue, action id 78: find a Fridge, action id
    79: find a Sink, action id 80: pick up the KeyChain, action
    id 81: pick up the Potato, action id 82: pick up the Pot,
    action id 83: pick up the Pen, action id 84: pick up the
    Candle, action id 85: pick up the CD, action id 86: pick up
    the Pan, action id 87: pick up the Watch, action id 88: pick
     up the Newspaper, action id 89: pick up the HandTowel,
    action id 90: pick up the SprayBottle, action id 91: pick up
     the BaseballBat, action id 92: pick up the Bread, action id
     93: pick up the CellPhone, action id 94: pick up the Book,
    action id 95: pick up the Lettuce, action id 96: pick up the
     CreditCard, action id 97: pick up the Mug, action id 98:
    pick up the AlarmClock, action id 99: pick up the Kettle,
    action id 100: pick up the ToiletPaper

> **EB-Habitat Action list**
>
> action id 0: navigate to the cabinet 7, action id 1: navigate
> to the cabinet 6, action id 2: navigate to the cabinet 5,
> action id 3: navigate to the cabinet 4, action id 4:
> navigate to the refrigerator push point, action id 5:
> navigate to the chair 1, action id 6: navigate to the table
> 1, action id 7: navigate to the table 2, action id 8:
> navigate to the TV stand, action id 9: navigate to the sink
> in the kitchen, action id 10: navigate to the right counter
> in the kitchen, action id 11: navigate to the left counter
> in the kitchen, action id 12: navigate to the sofa, action
> id 13: navigate to the refrigerator, action id 14: navigate
> to the left drawer of the kitchen counter, action id 15:
> navigate to the right drawer of the kitchen counter, action
> id 16: pick up the ball, action id 17: pick up the clamp,
> action id 18: pick up the hammer, action id 19: pick up the
> screwdriver, action id 20: pick up the padlock, action id
> 21: pick up the scissors, action id 22: pick up the block,
> action id 23: pick up the drill, action id 24: pick up the
> spatula, action id 25: pick up the knife, action id 26: pick
>  up the spoon, action id 27: pick up the plate, action id
> 28: pick up the sponge, action id 29: pick up the cleanser,
> action id 30: pick up the plum, action id 31: pick up the
> pear, action id 32: pick up the peach, action id 33: pick up
>  the apple, action id 34: pick up the lemon, action id 35:
> pick up the can, action id 36: pick up the box, action id
> 37: pick up the banana, action id 38: pick up the strawberry
> , action id 39: pick up the lego, action id 40: pick up the
> rubriks cube, action id 41: pick up the book, action id 42:
> pick up the bowl, action id 43: pick up the cup, action id
> 44: pick up the mug, action id 45: pick up the orange,
> action id 46: pick up the lid, action id 47: pick up the toy
>  airplane, action id 48: pick up the wrench, action id 49:
> place at the chair 1, action id 50: place at the table 1,
> action id 51: place at the table 2, action id 52: place at
> the TV stand, action id 53: place at the sink in the kitchen
> , action id 54: place at the right counter in the kitchen,
> action id 55: place at the left counter in the kitchen,
> action id 56: place at the sofa, action id 57: place at the
> refrigerator, action id 58: place at the left drawer of the
> kitchen counter, action id 59: place at the right drawer of
> the kitchen counter, action id 60: open the refrigerator,
> action id 61: close the refrigerator, action id 62: open the
>  cabinet 7, action id 63: open the cabinet 6, action id 64:
> open the cabinet 5, action id 65: open the cabinet 4, action
>  id 66: close the cabinet 7, action id 67: close the cabinet
>  6, action id 68: close the cabinet 5, action id 69: close
> the cabinet 4

**Part of Our RFT Action list**

action id 1: goto apple, action id 2: goto armchair, action id
   3: goto baseballbat, action id 4: goto basketball, action id
    5: goto bathtubbasin, action id 6: goto bed, action id 7:
   goto bowl, action id 8: goto box, action id 9: goto bread,
   action id 10: goto butterknife, action id 11: goto cabinet,
   action id 12: goto candle, action id 13: goto cart, action
   id 14: goto cellphone, action id 15: goto cloth, action id
   16: goto coffeemachine, action id 17: goto coffeetable,
   action id 18: goto countertop, action id 19: goto creditcard
   , action id 20: goto cup, action id 21: goto desk, action id
    22: goto desklamp, action id 23: goto diningtable, action
   id 24: goto dishsponge, action id 25: goto drawer, action id
    26: goto dresser, action id 27: goto egg, action id 28:
   goto floorlamp, action id 29: goto fork, action id 30: goto
   fridge, action id 31: goto garbagecan, action id 32: goto
   handtowelholder, action id 33: goto keychain, action id 34:
   goto knife, action id 35: goto laptop, action id 36: goto
   lettuce, action id 37: goto microwave, action id 38: goto
   mug, action id 39: goto newspaper, action id 40: goto
   ottoman, action id 41: goto pan, action id 42: goto pen,
   action id 43: goto pencil, action id 44: goto plate, action
   id 45: goto plunger, action id 46: goto pot, action id 47:
   goto potato, action id 48: goto remotecontrol, action id 49:
    goto safe, action id 50: goto shelf, action id 51: goto
   sidetable, action id 52: goto sinkbasin, action id 53: goto
   soapbar, action id 54: goto soapbottle, action id 55: goto
   sofa, action id 56: goto spatula, action id 57: goto spoon,
   action id 58: goto statue, action id 59: goto stoveburner,
   action id 60: goto tennisracket, action id 61: goto
   tissuebox, action id 62: goto toilet, action id 63: goto
   toiletpaper, action id 64: goto toiletpaperhanger, action id
    65: goto tomato, action id 66: goto vase, action id 67:
   goto watch, action id 68: goto wateringcan, action id 69:
   pickup alarmclock, action id 70: pickup apple, action id 71:
    pickup baseballbat, action id 72: pickup basketball, action
    id 73: pickup book, action id 74: pickup bowl, action id
   75: pickup box, action id 76: pickup bread, action id 77:
   pickup butterknife, action id 78: pickup candle, action id
   79: pickup cd, action id 80: pickup cellphone, action id 81:
    pickup cloth, action id 82: pickup creditcard, action id
   83: pickup cup, action id 84: pickup dishsponge, action id
   85: pickup egg, action id 86: pickup fork, action id 87:
   pickup glassbottle, action id 88: pickup handtowel, action
   id 89: pickup kettle, action id 90: pickup keychain, action
   id 91: pickup knife, action id 92: pickup ladle, action id
   93: pickup laptop, action id 94: pickup lettuce, action id
   95: pickup mug, action id 96: pickup newspaper, action id
   97: pickup pan, action id 98: pickup pen, action id 99:
   pickup pencil, action id 100: pickup peppershaker,

