# OpenReview forum: "On-policy Reinforcement Fine-tuning with Offline reward for Multi-step Embodied Planning"
_ICLR.cc/2026/Conference — ICLR 2026 Conference Withdrawn Submission_

### Official Review · Reviewer_yvuV · 2025-10-30

**Soundness:** 2
**Presentation:** 2
**Contribution:** 2
**Rating:** 4
**Confidence:** 4

**Summary:**

This paper proposes an on-policy Reinforcement Fine-Tuning (RFT) framework for multi-step embodied planning. To address the challenges of costly online interaction and sparse rewards, the method utilizes an offline reward signal computed by comparing model-generated action sequences to expert demonstrations, primarily based on the length of their correct action prefix. The approach involves an initial SFT stage, followed by policy optimization using GRPO. Experimental evaluation on the Embench benchmark is conducted to demonstrate the effectiveness of the proposed pipeline.

**Strengths:**

The paper presents a valuable application of Reinforcement Fine-Tuning to the embodied planning domain. The designed reward function, which combines a prefix-based accuracy reward with a structured format reward, successfully adapts the RFT paradigm to this task and shows a clear performance improvement over SFT alone, as evidenced in the ablation studies.

**Weaknesses:**

- The proposed method's core reward mechanism is fundamentally dependent on pre-defined expert trajectories. This strong reliance on supervised demonstration data limits the agent's capacity for strategic exploration and discovery of novel solutions beyond the provided examples, constraining its generalization to new scenarios.

- Insufficient Baseline Comparisons. The experimental section lacks comparisons with previous baselines specifically designed for embodied task planning. For instance, the method is not compared against DPO-based methods like EPO, D2PO, SFT-based planners like WAP (62.7%), or other RL/RFT-based VLM methods such as RL4VLM (51.2%), VAGEN (52.8%).

- There is a critical conceptual flaw in the training setup. The paper studies a multi-turn, interactive problem but explicitly avoids "online execution" and "collecting interactive feedback" (Line 142). Consequently, during training, the model's rollouts are generated from the sequence of states in the expert trajectory, not from the states resulting from the model's own previous actions. This severs the feedback loop that is central to sequential decision-making, as the state at step_{i+1} is independent of the action taken at step_i. This reduces the problem from interactive Sequential Decision-Making to a static Sequence Generation task, failing to address the core challenge of adapting to dynamic state transitions.

- The entire study is conducted using a single backbone model, it is important to demonstrate the generality of the proposed RFT framework across different model architectures.

**Questions:**

- The EB-ALFRED benchmark and the proposed method's training data are both derived from the ALFRED dataset. Is there any overlap between the tasks or trajectories used for RFT training and those in the Embench evaluation sets?

- The main results table omits the "Long Horizon" subset of Embench. Could you provide the results on this subset? Long Horizon capability is also important in embodied planning.

-  What's the backbone model used for SFT and RFT? The main text does not explicitly state this.

---

> ### Author Response · Authors · 2025-11-26
> **Response (1/4)**
>
> # W1： Expert Reliance vs. Exploration and Generalization
>
> > W1：The proposed method's core reward mechanism is fundamentally dependent on pre-defined expert trajectories. This strong reliance on supervised demonstration data limits the agent's capacity for strategic exploration and discovery of novel solutions beyond the provided examples, constraining its generalization to new scenarios.
>
> We acknowledge the reviewer’s concern that relying on expert trajectories might theoretically limit the discovery of novel solutions compared to open-ended exploration. However, we argue that in the context of embodied planning, **stability and dense feedback often outweigh the theoretical benefits of sparse-reward exploration**, especially given the practical constraints of real-world deployment.
>
> Crucially, empirical results demonstrate that our method does not merely "cope" with this limitation but actually **outperforms** state-of-the-art Online RL methods. Online approaches, while capable of exploration, often struggle with **reward sparsity** in long-horizon tasks, leading to inefficient learning. In contrast, our dense, prefix-based offline reward provides a robust learning signal that drives superior generalization.
>
> To validate this, we compared our method (using the same augmented SFT backbone with era[1]) against leading Online RL baselines on **EB-ALFRED**.
>
> | **Method**          | Ours (offline reward)  | **ERA (Online reward)** |
> | ------------------- | -------------------------- | ----------------------- |
> | **EB-ALFRED (Avg)** | **70.88**                  | 65.2                    |
>
> To further support our point, we will talk about more experiment details and baselines in the response to W2
>
>
>
> # W2：Insufficient Baseline Comparisons
>
> > W2:The experimental section lacks comparisons with previous baselines specifically designed for embodied task planning.
>
> We thank the reviewer for suggesting a comparison with more baselines like Online RL baselines . To provide a rigorous and informative comparison, we conducted additional experiments against several state-of-the-art methods evaluated on EmbodiedBench.
>
> **1. Baselines and Experimental Setup**
>
> We selected the following representative and recent baselines that utilize online RL:
>
> - **ERA**[1]:  A state-of-the-art Online RL framework that integrates Embodied Prior Learning (EPL) with **turn-level** PPO optimization to handle embodied task planining.
>
> - **RL4VLM**[2] & **VAGEN**[3]: Also employ Online RL optimization strategies. Specifically, **RL4VLM** applies standard **token-level** PPO to VLMs, while **VAGEN** employs a **bi-level** advantage estimation strategy to stabilize online training.
>
> - **WAP**[4]:  Utilizes extensive Data Augmentation for robust SFT.
>
>
>
> The experimental results for **ERA, RL4VLM, and VAGEN** are directly cited from the era paper[1] (October 2025), where these methods were evaluated under a unified setting on EmbodiedBench.
>
> A critical observation is that these baselines utilize heavy data augmentation during their SFT stage, whereas our original method relied on a smaller, unaugmented dataset. This created a performance disparity in the base SFT models. **To ensure an fair comparison of the optimization methods** (Ours RL vs. Online RL), we re-trained our SFT backbone using the same augmented dataset used by these baselines. This places all methods on the same starting line. We then applied our RFT with offline dataset on top of this enhanced SFT model.
>
>
>
> **2. Experimental Results**
>
> The results on the **EB-ALFRED** benchmark are summarized below:
>
> | **Method**          | Ours (with augmented SFT) | Only augmented SFT | ERA  | **WAP** | **RL4VLM** | **VAGEN** |
> | ------------------- | ------------------------------ | ------------------ | -------- | ------- | ---------- | --------- |
> | **EB-ALFRED (Avg)** | **70.88**                      | 56.0               | 65.2     | 62.7    | 51.2       | 52.8      |
>
> Our method (70.88%) outperforms the strong Online RL baseline ERA (65.2%) and significantly surpasses RL4VLM and VAGEN. This validates that our **dense, prefix-based offline reward** effectively mitigates the sparse reward problem inherent in online embodied planning, providing a more stable learning signal than sparse success/failure feedback.
>
> In summary, our approach not only avoids the high computational cost of online interaction but also delivers superior performance compared to standard Online RL methods in this domain.
>
>
>
> [1] Era: Transforming vlms into embodied agents via embodied prior learning and online reinforcement learning." arXiv:2510.12693
>
> [2]Fine-tuning large vision-language models as decision-making agents via reinforcement learning, Neurips 2024
>
> [3] VAGEN: Reinforcing world model reasoning for multi-turn vlm agents. arXiv:2510.16907
>
> [4] World-aware Planning Narratives Enhance Large Vision-Language Model Planner." arXiv:2506.21230

---

> ### Author Response · Authors · 2025-11-26
> **Response (2/4)**
>
> # W3：Conceptual Flaw regarding Static vs. Interactive Training
>
> > W3: There is a critical conceptual flaw in the training setup. The paper studies a multi-turn, interactive problem but explicitly avoids "online execution" and "collecting interactive feedback" (Line 142). This reduces the problem from interactive Sequential Decision-Making to a static Sequence Generation task, failing to address the core challenge of adapting to dynamic state transitions.
>
> We thank the reviewer for this sharp observation. We clarify that our approach deliberately decouples the **training formalism** from the **evaluation protocol** to address the efficiency and sparsity bottlenecks of embodied learning. While the training phase is indeed formulated as a Single-Step MDP, our extensive empirical results demonstrate that this effectively translates to improved performance in the fully interactive, multi-turn evaluation setting.
>
> * **Training Phase (Single-Step MDP / Contextual Bandit):**
>   As described in the "Offline Expert Trajectory Construction" (L268-271) and similar to standard RLHF (e.g., PPO/GRPO for LLMs), we formulate training as a **single-step MDP** (or Contextual Bandit).
>
>   * **State ($s$):** A composite input containing the task instruction $g$, the history of actions $a_{0:n-1}$, and the visual observation $o_n$ recorded in the expert dataset.
>   * **Action ($a$):** The model predicts the **entire remaining sequence** of actions $\hat{a}_{n:} = \{a_n, \dots, a_k\}$.
>   * **Transition & Observation:** Since the model outputs the full future plan in one turn relative to the provided state, the episode ends immediately after the prediction. No new observation $o_{n+1}$ is needed from a simulator because the reward is computed by comparing the predicted plan against the offline expert ground truth.
>   * **Reward:** The reward is computed offline based on the correctness of the generated sequence relative to the expert trajectory.
>
> * **Evaluation Phase: Multi-Turn Dynamic MDP**：
>
>   Crucially, during evaluation (Section 3.1), we strictly follow the **EmbodiedBench** protocol . This is a fully interactive, **Multi-Turn Dynamic MDP** where:
>
>   - The agent predicts actions step-by-step (or in chunks).
>   - These actions are executed in the simulator (AI2-THOR/Habitat), generating actual state transitions and new visual observations.
>   - If an action fails or is invalid, the agent receives feedback and must replan based on the updated state.
>
> In conclusion, although the training phase abstracts away the immediate dynamic transitions to enable efficient, large-scale optimization, the learned policy successfully generalizes to the dynamic setting. By optimizing the model to generate long-horizon expert-aligned trajectories given any intermediate state, we effectively improve its ability to handle the sequential decision-making challenges inherent in the multi-turn evaluation.

---

> ### Author Response · Authors · 2025-11-26
> **Response (3/4)**
>
> # W4&Q3：backbone
>
> > W4：The entire study is conducted using a single backbone model, it is important to demonstrate the generality of the proposed RFT framework across different model architectures.
> >
> > Q3：What's the backbone model used for SFT and RFT? The main text does not explicitly state this.
>
> We thank the reviewer for the suggestion to verify the generality of our framework.
>
> **1.Clarification of the Primary Backbone (Q3)**
>
> We confirm that the backbone model used for all main experiments in the paper is Qwen2.5-VL-7B. While this was documented in Appendix E.2 2, we acknowledge it was not sufficiently prominent in the main text. We will explicitly state this in Section 3.1 of the revised manuscript.
>
> **2.Generalization to InternVL2.5-8B (W4)**
>
> To demonstrate that our On-policy RFT with Offline Reward framework is model-agnostic, we conducted additional experiments using a completely different architecture: **InternVL2.5-8B.** We applied the exact same pipeline (SFT $\to$ Offline RFT) to this new backbone.
>
> **Note on Experimental Setup:** Due to the time constraints of the rebuttal phase, we conducted this specific generalization experiment using our **standard (non-augmented) SFT dataset**. Consequently, the absolute performance scores reported below differ from the "Augmented SFT" results presented in **the Response to W2**. However, the key metric for this comparison is the *relative improvement* of RFT over the SFT baseline, which isolates the contribution of our reinforcement fine-tuning method.
>
> | **Backbone**       | **Method**     | **EB-ALFRED (Avg)** | **Base** | **Com** | **Cplx** | **Visual** | **Spatial** |
> | ------------------ | -------------- | ------------------- | -------- | ------- | -------- | ---------- | ----------- |
> | **Qwen2.5-VL-7B**  | Base           | 2                   | 4        | 2       | 2        | 2          | 0           |
> |                    | SFT            | 22                  | 34       | 22      | 24       | 12         | 18          |
> |                    | **Ours (RFT)** | 49.2                | 60       | 60      | 48       | 38         | 40          |
> | **InternVL2.5-8B** | Base           | 3.6                 | 2        | 0       | 12       | 0          | 4           |
> |                    | SFT            | 28                  | 36       | 28      | 32       | 22         | 22          |
> |                    | **Ours (RFT)** | 54.4                | 60       | 66      | 52       | 44         | 48          |

---

> ### Author Response · Authors · 2025-11-26
> **Response (4/4)**
>
> # Q1：Dataset Overlap
>
> > Q1：The EB-ALFRED benchmark and the proposed method's training data are both derived from the ALFRED dataset. Is there any overlap between the tasks or trajectories used for RFT training and those in the Embench evaluation sets?
>
> We confirm that our RFT training data is derived from the original **ALFRED** dataset. However, we emphasize that **Embench (EB-ALFRED)** is not merely a subset of ALFRED; it introduces significant distributional shifts that minimize direct overlap and prevent trivial memorization:
>
> - **Action Space Modification:** Embench dynamically adjusts the action space (ranging from 171 to 298 actions) and merges redundant primitives (e.g., unified "put down").
> - **Task & Object Diversity:** It introduces object instance variations and modifies task specifications to be more robust and realistic compared to the original templates.
>
> **Crucially, to rigorously test generalization beyond potential dataset similarities**, we evaluated our model on **EB-Habitat** (Table 1, Right). EB-Habitat represents a **completely out-of-domain environment** with distinct scenes, object assets, physics engines, and action definitions . Our strong performance on EB-Habitat serves as robust evidence that the learned planning policy generalizes well to unseen embodied environments.
>
>
>
>
>
> # Q2：Performance on Long-Horizon Tasks
>
> > Q2：The main results table omits the "Long Horizon" subset of Embench. Could you provide the results on this subset? Long Horizon capability is also important in embodied planning.
>
> We thank the reviewer for noting the importance of long-horizon capabilities. Due to space constraints in the main text, the full results for the "Long Horizon" subset were reported in the Appendix Table 6
>
> **Results on Long-Horizon Subset:**
>
> | **Method**       | **Base Model** | **SFT Model** | **Ours (RFT)** |
> | ---------------- | -------------- | ------------- | -------------- |
> | **Success Rate** | 6.0%           | 8.0%          | **24.5%**      |
>
> While our method achieves a clear improvement (**24.5%**) over both the Base (**6%**) and SFT (**8%**) baselines, we acknowledge that the absolute gain is more modest compared to other task categories.
>
> While we agree with the reviewer’s interest in solving ultra-long-horizon challenges, we would like to clarify that this is not the central claim of our current work, which focuses on establishing a scalable, efficient offline reinforcement framework. We view solving the specific complexities of extremely long horizons as an important next step built on this foundation, and we plan to explore architectures specifically designed for long-term memory in future work.

---

### Official Review · Reviewer_TFDL · 2025-11-02

**Soundness:** 3
**Presentation:** 3
**Contribution:** 4
**Rating:** 6
**Confidence:** 4

**Summary:**

This paper tackles training VLMs in embodied environments, where sparse rewards and costly simulator rollouts make RL difficult. The authors propose an *on-policy* RFT method that uses *offline rewards* computed by comparing model-generated trajectories to expert demonstrations, avoiding expensive environment interaction. The approach combines supervised imitation learning with offline RFT and provides theoretical guarantees that the learned policy approximates the true online optimum. Experiments on embodied reasoning benchmarks show substantial gains over strong open- and closed-source baselines, indicating improved planning and generalization.

**Strengths:**

- The theoretical analysis is clear and shows that the offline objective is directional aligned with the true reward. This mathematical justification is appreciated rather than solely relying on empirical intuition.
- The idea is intuitively motivated and well executed. Avoiding the cost of simulator rollouts while retaining the performance is impressive.
- By eliminating the need for rollouts, the approach makes the path towards scalable training more practical and accessible.

**Weaknesses:**

- The method's reliance on offline expert trajectories means it cannot adapt when the environment, task structure, or available actions differ significantly from those seen in training. All experiments are conducted within variations of similar embodied reasoning environments (ALFRED and Habitat).
- On the same note, the learned policy assumes a shared task schema and action grammar; if either changes, the offline reward function provides no meaningful signal.
- The paper repeatedly emphasizes computational and practical efficiency as a key motivation, but never provides concrete evidence (e.g., training time, resource usage, or sample complexity) to support this claim.
- While the paper demonstrates improvements from the combined SFT→RFT pipeline and compares several internal reward variants, it never evaluates a control setup that performs *on-policy fine-tuning without the proposed offline reward*. A fairer test would include something like "SFT → on-policy fine-tuning without offline reward" versus "SFT → on-policy fine-tuning with offline reward." Comparing the offline reward signal against the true environment reward (when available) would reveal how closely it approximates real task success and how much performance is lost or retained relative to genuine RL feedback.

**Questions:**

- Could the authors clarify whether their approach can be extended to environments with continuous actions or richer affordances, where the notion of a discrete prefix match no longer applies?
- How sensitive is the method to suboptimal or noisy expert trajectories?
- Given the offline reward’s limitations on unseen tasks, could the authors comment on the feasibility of combining a small number of online rollouts with their offline reward for fine-tuning?

---

> ### Author Response · Authors · 2025-11-26
> **Response (1/5)**
>
> # Q1, W1 ,W2: Extensibility to Continuous Actions and Diverse Environments
>
> > Q1:Could the authors clarify whether their approach can be extended to environments with continuous actions or richer affordances, where the notion of a discrete prefix match no longer applies?
> >
> > W1&W2: The method's reliance on offline expert trajectories means it cannot adapt when the environment, task structure, or available actions differ significantly from those seen in training.On the same note, the learned policy assumes a shared task schema and action grammar
>
> We acknowledge that the specific **prefix-matching** reward formulation used in our main experiments is tailored for discrete action spaces. However, we emphasize that the core framework of **On-policy RFT with Offline Rewards** is highly agnostic to the specific reward metric and can be readily extended to continuous environments or richer affordances.
>
> **1.Extensibility via Metric-Based Rewards**
>
> As demonstrated in our Ablation Study (Section 3.3), we explicitly evaluated alternative reward formulations that **do not rely on discrete token** matching, specifically Embedding Cosine Similarity and Optimal Transport (OT) Distance. These metric-based rewards are naturally compatible with continuous action spaces
>
> Our results show that while the prefix-based reward was optimal for the discrete ALFRED benchmark, the distance-based `Embed-OT` reward still achieved a highly competitive success rate on EB-ALFRED. This confirms that the framework functions effectively even when the reward signal is derived from soft similarity metrics rather than discrete matching.
>
>
>
> **2.Alignment with Recent Paradigm Shifts**
>
> The paradigm of using offline expert similarity to drive on-policy optimization is proving to be highly generalizable. Notably, very recent works\[1][2] in the language domain have independently validated this approach, utilizing cosine similarity between model outputs and targets to guide reinforcement learning.
>
> Therefore, extending our method to continuous domains or new task schemas does not require changing the framework itself. It simply requires substituting the discrete prefix-check with an appropriate continuous distance metric (e.g., trajectory MSE or embedding similarity), making the approach robust and adaptable to diverse embodied settings.
>
>
>
> [1] Supervised Reinforcement Learning: From Expert Trajectories to Step-wise Reasoning, arXiv:2510.25992
>
> [2] "On-Policy Distillation", Thinking Machines Lab, https://thinkingmachines.ai/blog/on-policy-distillation

---

> ### Author Response · Authors · 2025-11-26
> **Response (2/5)**
>
> # W3：Limited discussion of computational efficiency.
>
> > W3: The paper repeatedly emphasizes computational and practical efficiency as a key motivation, but never provides concrete evidence (e.g., training time, resource usage, or sample complexity) to support this claim.
>
> We thank the reviewer for raising this important point regarding computational efficiency.
>
> To address the request for quantitative evidence, we conducted a direct comparison between our **Offline RFT** and a standard **Online RL** setup (based on the ERA[1] framework). We measured the training time per update step, total convergence time, and final performance on the EB-ALFRED benchmark. All experiments were conducted on 8x H200 GPUs, though our method is deployable on a minimum configuration of 4x 80G VRAM.
>
> **Table: Comparison of Training Efficiency and Performance**
>
> | **Metric**                  | **Ours (Offline RFT)**                                       | **Online RL (Baseline)**                                     | **Comparison**        |
> | --------------------------- | ------------------------------------------------------------ | ------------------------------------------------------------ | --------------------- |
> | **Time per Update Step**    | **~2 min**                                                   | 12 - 15 min                                                  | **~6x - 7.5x Faster** |
> | **Total Training Time**     | **8 - 10 hours**                                             | 36 - 40 hours                                                | **~4x Faster**        |
> | **Final Performance (Avg)** | **70.88%**                                                   | 65.2%                                                        | **+5.68% Higher**     |
> | **Environment Setup**       | **Standard VLM+RL**  (e.g., OpenRLHF, VeRL).  <br />Simple to deploy. | **Complex Integration**  (Simulator + Headless Rendering/Xorg + VLM-RL Framework). <br /> High engineering barrier. | **Plug-and-Play**     |
>
> As shown in the table, the Online RL method suffers from significant latency due to multi-step simulator rendering, environment resets, and the synchronization overhead between the VLM inference and the physics engine (taking 12-15 mins per step). In contrast, our method decouples the environment, treating the task as efficient text generation with fast string-matching rewards, reducing the step time to just **2 minutes**. Crucially, this efficiency does not come at the cost of performance. Our method achieves a higher success rate (**70.88%**) in significantly less time (**8-10 hours** vs. 36-40 hours).
>
> Finally, it is crucial to note that while simulators *can* be parallelized, **real-world online training** is prohibitively expensive and often infeasible due to safety risks and high reset costs. Our method, by leveraging offline data for policy optimization, avoids these physical risks entirely, offering a safer and more scalable path for real-world RL.
>
>
>
> [1] Era: Transforming vlms into embodied agents via embodied prior learning and online reinforcement learning." arXiv:2510.12693

---

> ### Author Response · Authors · 2025-11-26
> **Response (3/5)**
>
> # W4: Comparison with Online RL Baselines
>
> > W4: While the paper demonstrates improvements from the combined SFT→RFT pipeline and compares several internal reward variants, it never evaluates a control setup that performs *on-policy fine-tuning without the proposed offline reward*. Comparing the offline reward signal against the true environment reward (when available) would reveal how closely it approximates real task success and how much performance is lost or retained relative to genuine RL feedback.
>
> We thank the reviewer for suggesting a comparison with Online RL baselines. To provide a rigorous and informative comparison, we conducted additional experiments against several state-of-the-art Online RL methods evaluated on EmbodiedBench.
>
> **1. Baselines and Experimental Setup**
>
> We selected the following representative and recent baselines that utilize online RL:
>
> - **ERA**[1]:  A state-of-the-art Online RL framework that integrates Embodied Prior Learning (EPL) with **turn-level** PPO optimization to handle embodied task planining.
>
> - **RL4VLM**[2] & **VAGEN**[3]: Also employ Online RL optimization strategies. Specifically, **RL4VLM** applies standard **token-level** PPO to VLMs, while **VAGEN** employs a **bi-level** advantage estimation strategy to stabilize online training.
>
> - **WAP**[4]:  Utilizes extensive Data Augmentation for robust SFT.
>
>
>
> The experimental results for **ERA, RL4VLM, and VAGEN** are directly cited from the era paper[1] (October 2025), where these methods were evaluated under a unified setting on EmbodiedBench.
>
> A critical observation is that these baselines utilize heavy data augmentation during their SFT stage, whereas our original method relied on a smaller, unaugmented dataset. This created a performance disparity in the base SFT models. **To ensure an fair comparison of the optimization methods** (Ours RL vs. Online RL), we re-trained our SFT backbone using the same augmented dataset used by these baselines. This places all methods on the same starting line. We then applied our RFT with offline dataset on top of this enhanced SFT model.
>
>
>
> **2. Experimental Results**
>
> The results on the **EB-ALFRED** benchmark are summarized below:
>
> | **Method**          | **Ours (with augmented SFT) ** | Only augmented SFT | **ERA ** | **WAP** | **RL4VLM** | **VAGEN** |
> | ------------------- | ------------------------------ | ------------------ | -------- | ------- | ---------- | --------- |
> | **EB-ALFRED (Avg)** | **70.88**                      | 56.0               | 65.2     | 62.7    | 51.2       | 52.8      |
>
> Our method (70.88%) outperforms the strong Online RL baseline ERA (65.2%) and significantly surpasses RL4VLM and VAGEN. This validates that our **dense, prefix-based offline reward** effectively mitigates the sparse reward problem inherent in online embodied planning, providing a more stable learning signal than sparse success/failure feedback.
>
> In summary, our approach not only avoids the high computational cost of online interaction but also delivers superior performance compared to standard Online RL methods in this domain.
>
>
>
> [1] Era: Transforming vlms into embodied agents via embodied prior learning and online reinforcement learning." arXiv:2510.12693
>
> [2]Fine-tuning large vision-language models as decision-making agents via reinforcement learning, Neurips 2024
>
> [3] VAGEN: Reinforcing world model reasoning for multi-turn vlm agents. arXiv:2510.16907
>
> [4] World-aware Planning Narratives Enhance Large Vision-Language Model Planner." arXiv:2506.21230

---

> ### Author Response · Authors · 2025-11-26
> **Response (4/5)**
>
> # Q2：Robustness to Sub-Optimal Trajectories
>
> > Q2: How sensitive is the method to suboptimal or noisy expert trajectories?
>
> We thank the reviewer for raising this practical concern regarding the availability and quality of expert data. We address this from both theoretical and empirical perspectives.
>
> **1.Theoretical Perspective: Optimality is Not Strict**
>
> In embodied planning, "expert" trajectories are rarely uniquely optimal; a task can often be solved via multiple valid action sequences. Our theoretical analysis in Section 2.3.2 (Proposition 1) anticipates this. We prove that the policy optimization remains directionally correct and bounded relative to the true environment objective, even if the reference trajectory is just one of many valid solutions. The reward signal $R_{accuracy}$ primarily encourages alignment with successful behavior patterns rather than strict memorization of a single "golden" path.
>
> **2.Empirical Evidence: Perturbation Study**
>
> To empirically quantify robustness against sub-optimal data, we conducted a controlled experiment where we intentionally degraded the quality of our RFT training dataset. We introduced noise into the expert trajectories using two perturbation strategies:
>
> - **Insertion of Irrelevant Actions:** Randomly injecting non-functional actions into the sequence.
> - **Redundant Loops:** Inserting closed-loop navigation paths (e.g., moving away and returning) to create inefficient trajectories.
>
> We trained our model using datasets with **10%** and **20%** of the trajectories perturbed, respectively. The results on the EB-ALFRED benchmark (using our augmented SFT backbone for RFT) are shown below:
>
> | **RFT Data Quality**           | **Avg**   | **Base** | **Com** | **Cplx** | **Visual** | **Spatial** |
> | ------------------------------ | --------- | -------- | ------- | -------- | ---------- | ----------- |
> | **0% Perturbation (Standard)** | **70.88** | 76.4     | 64      | 82       | 72         | 60          |
> | **10% Perturbation**           | **68.88** | 76.4     | 64      | 76       | 68         | 60          |
> | **20% Perturbation**           | 62.50     | 74.5     | 52      | 76       | 62         | 48          |
>
> - **High Resilience at 10% Noise:** Even with 10% of the training data corrupted by sub-optimal noise, the model maintains an impressive average success rate of **68.88%**. Notably, this still outperforms the state-of-the-art proprietary model **Claude-3.5-Sonnet (66.4%)**.
> - **Graceful Degradation at 20% Noise:** At 20% perturbation, while performance drops to 62.5%, the model remains highly competitive, significantly outperforming the base GPT-4o model (56.8%).
>
> This confirms that our method does not require "perfect" expert data to succeed and can robustly extract planning policies even from datasets containing a significant portion of sub-optimal or noisy trajectories.

---

> ### Author Response · Authors · 2025-11-26
> **Response (5/5)**
>
> # Q3：Feasibility of Combining Online Rollouts with Offline Rewards
>
> > Q3: Given the offline reward’s limitations on unseen tasks, could the authors comment on the feasibility of combining a small number of online rollouts with their offline reward for fine-tuning?
>
> We appreciate the reviewer’s insightful suggestion. While combining online and offline signals is theoretically ideal, we would like to highlight the engineering context and future potential of this direction.
>
> **1. Engineering Context at the Time of Research** Implementing Online RL for Embodied VLMs is a significant engineering challenge, primarily due to the complexity of synchronizing massive model inference with simulator environments. At the time of our research, efficient frameworks capable of seamlessly coupling Embodied Simulators with VLA /VLM training were not available. Mature frameworks for this specific integration, such as **SimpleVLA-RL** [1], **RLinf** [2], and **ERA** [3], have only begun to emerge very recently (e.g., September 2025), which is subsequent to the completion of our work.
>
> **2. Future Feasibility and Hybrid Approaches** With the advent of these recent VLM+Simulator training frameworks, combining online and offline rewards is becoming increasingly feasible. We believe this is a compelling direction for future work. For instance, a hybrid approach could utilize successful trajectories discovered during online exploration to dynamically expand the "expert" dataset used for our offline reward computation. This would allow the system to bootstrap its own experience into the stable, offline optimization loop we have established.
>
>
>
> [1] "Simplevla-rl: Scaling vla training via reinforcement learning." arXiv:2509.09674.
>
> [2] "RLinf-VLA: A Unified and Efficient Framework for VLA+ RL Training."  arXiv:2510.06710.
>
> [3] Era: Transforming vlms into embodied agents via embodied prior learning and online reinforcement learning." arXiv:2510.12693

---

### Official Review · Reviewer_9WSH · 2025-11-03

**Soundness:** 2
**Presentation:** 1
**Contribution:** 2
**Rating:** 2
**Confidence:** 4

**Summary:**

The authors present a method for fine-tuning Vision-Language Models (VLMs) for embodied agent planning using reinforcement learning. The primary contribution is an offline reward mechanism designed to overcome the challenges of sparse rewards and costly simulation. This reward is calculated by comparing model-generated rollouts to expert trajectories. Experiments conducted on the Embench benchmark demonstrate that this approach achieves better performance compared to the original base models, Supervised Fine-Tuning (SFT) models, and several open-source LLMs.

**Strengths:**

1. The paper is well-motivated. It aims to address an important problem: the sparse rewards and costly simulation for online RL embodied AI training.

2. The experimental results show it outperforms the base model and the SFT model by a big margin on both EB-ALFRED and EB-Habitat.

**Weaknesses:**

1. The reviewer found many technique details are not presented clearly and needs further clarification. A critical point requiring clarification is the precise mechanism for generating the "model rollouts." The authors state that the method aims to address costly simulation, yet it is unclear whether these rollouts require interaction with a simulator. This ambiguity makes it difficult for the reader to fully understand the process.

- If no simulator is used: The manuscript should provide a detailed explanation of how these rollouts are performed (e.g., auto-regressively by the model). Furthermore, it is unclear how the method maintains or approximates an "on-policy" data distribution in the absence of an interactive environment.

- If a simulator is used: This appears to contradict the paper's core motivation of avoiding high simulation costs. The authors should explicitly address this potential contradiction and clarify the computational costs associated with this step, explaining how it is less "costly" than traditional online methods.

A more explicit description of this rollout process is essential for understanding the method's novelty and practical advantages.

2. In Line 82, the authors claim their work is the first to apply RL to optimize VLM for embodied tasks. This claim appears to be an overstatement, as several existing works utilize RL for similar tasks. For example, [a] employs both Supervised Fine-Tuning (SFT) and online RL to fine-tune embodied agents, demonstrating strong generalization performance across diverse benchmarks. A comparison with [a] in the experimental section would significantly enhance the informativeness of the paper.

3. The description of the data filtering process in Lines 190-193 lacks the necessary precision. The manuscript uses ambiguous terms such as 'too few' and 'too many' without providing formal, quantitative definitions. To ensure the method is reproducible, the authors must specify the exact numerical thresholds or statistical criteria used to implement this filtering step.


[a] From Multimodal LLMs to Generalist Embodied Agents: Methods and Lessons, Szot et al., CVPR’25

**Questions:**

1. The proposed approach relies on expert trajectories for reward computation. How robust is this method if, as is often the case in real-world scenarios, expert data is expensive or unavailable, and only sub-optimal trajectories can be obtained?

2. Could you clarify the meaning of the terms k+1 and n+1 as they appear in equation (3)?

3. How are out-of-distribution issues addressed in the reward computation? Specifically, when the policy diverges from the expert policy, do you assume  it always leads to suboptimal actions?

4. The results in Table 1 are presented without confidence intervals (CIs). Given that reinforcement learning methods are known to have high variance, CIs are essential for a statistically robust evaluation and for making reliable comparisons between approaches. Please add confidence intervals (or standard deviations) for all results and specify the number of runs (e.g., random seeds) used to compute these statistics.

5. Please compare with more RL baselines such as [a]

---

> ### Author Response · Authors · 2025-11-26
> **Response (1/5)**
>
> # W1：Clarification on "Model Rollout" Mechanism
>
> > W1: The reviewer found many technique details are not presented clearly and needs further clarification. A critical point requiring clarification is the precise mechanism for generating the "model rollouts." The authors state that the method aims to address costly simulation, yet it is unclear whether these rollouts require interaction with a simulator. This ambiguity makes it difficult for the reader to fully understand the process.
>
> We thank the reviewer for highlighting this ambiguity. We explicitly confirm that **no simulator is used during the training rollout process**. This aligns with our core motivation to avoid high simulation costs. Below, we clarify the precise mechanism of the rollout and how "on-policy" learning is achieved without an interactive environment.
>
>
>
> **1.Clarification of the Underlying Decision Process**
>
> Our training phase does not involve a dynamic simulator transition function P(s'|s,a). Instead, we decouple the training and evaluation processes as follows:
>
> * **Training Phase (Single-Step MDP / Contextual Bandit):**
>   As described in the "Offline Expert Trajectory Construction" (L268-271) and similar to standard RLHF (e.g., PPO/GRPO for LLMs), we formulate training as a **single-step MDP** (or Contextual Bandit).
>
>   * **State ($s$):** A composite input containing the task instruction $g$, the history of actions $a_{0:n-1}$, and the visual observation $o_n$ recorded in the expert dataset.
>   * **Action ($a$):** The model predicts the **entire remaining sequence** of actions $\hat{a}_{n:} = \{a_n, \dots, a_k\}$.
>   * **Transition & Observation:** Since the model outputs the full future plan in one turn relative to the provided state, the episode ends immediately after the prediction. No new observation $o_{n+1}$ is needed from a simulator because the reward is computed by comparing the predicted plan against the offline expert ground truth.
>   * **Reward:** The reward is computed offline based on the correctness of the generated sequence relative to the expert trajectory.
>
> * **Evaluation Phase (Multi-Turn Dynamic MDP):**
>   During evaluation (Section 3.1), we strictly follow the **EmbodiedBench** protocol. This is a fully interactive, multi-turn MDP where the agent predicts actions step-by-step (or in chunks), executes them in the simulator (AI2-THOR/Habitat), and receives actual state transitions and new visual observations.
>
>
>
> **2.The Precise Rollout Mechanism**
>
> The "model rollout" in our framework is a standard auto-regressive generation process performed solely by the VLM, identical to inference in Large Language Models. It does not involve environment interaction.
>
> - **Input:** The model receives a static state $s$ from our offline dataset, which comprises:
>
>   1. The task instruction $g$.
>   2. The action history $a_{0:n-1}$ (text).
>   3. The visual observation $o_n$ (image) at step $n$.
>
> - **Generation:** Conditioned on $s$, the VLM auto-regressively generates the **entire sequence of future actions** $\hat{a}_{n:} = \{a_n, \dots, a_k\}$ as text tokens.
>
> - **Termination:** The generation stops when the model outputs an end-of-sequence token or reaches a maximum token limit.
>
> - **Evaluation:** Once generated, this sequence is immediately scored against the ground-truth expert trajectory stored in the dataset. No simulator step function is called.
>
>
>
> **3. Definition of "On-Policy"**
>
> > W1：Furthermore, it is unclear how the method maintains or approximates an "on-policy" data distribution in the absence of an interactive environment.
>
> In our framework, **"on-policy"** is defined as **sampling trajectories directly from the current policy $\pi_\theta$ for optimization**, rather than using a fixed data buffer or teaher-forcing method.
>
> * **Specifically:** We sample $a \sim \pi_\theta(\cdot|s)$. The model generates its own rollouts (plans) based on the input state. These rollouts are then scored, and the policy is updated to increase the probability of high-reward rollouts generated by the *current* model.
> * **Why this distinction matters:** This fundamentally differs from Supervised Fine-Tuning (SFT). As analyzed in Section 2.3.1 and corroborated by recent theoretical work [1-2] , **on-policy sampling** is critical for generalization and preventing forgetting. Unlike SFT, which imitates fixed data, our approach optimizes the *current* policy distribution, allowing the model to learn from self-generated negative examples and thereby achieving stronger robustness.
>
>
>
> [1] RL's Razor: Why Online Reinforcement Learning Forgets Less. arXiv:2509.04259.
>
> [2] Retaining by Doing: The Role of On-Policy Data in Mitigating Forgetting. arXiv:2510.18874

---

> ### Author Response · Authors · 2025-11-26
> **Response (2/5)**
>
> # W2/Q5: Comparison with Online RL Baselines
>
> > W2&Q5: A comparison with [1] in the experimental section would significantly enhance the informativeness of the paper.Please compare with more RL baselines such as [1]
>
> We thank the reviewer for suggesting a comparison with Online RL baselines, such as Szot et al. [1]. To provide a rigorous and informative comparison, we conducted additional experiments against several state-of-the-art Online RL methods evaluated on EmbodiedBench.
>
> **1. Baselines and Experimental Setup**
>
> Since Szot et al. [1] (Online PPO) was not originally benchmarked on Embench, we selected the following representative and recent baselines that utilize similar methodologies:
>
> - **ERA**[2]:  A state-of-the-art Online RL framework that integrates Embodied Prior Learning (EPL) with **turn-level** PPO optimization to handle embodied task planining.
> - **WAP**[3]:  Utilizes extensive Data Augmentation for robust SFT.
> - **RL4VLM**[4] & **VAGEN**[5]: Also employ Online RL optimization strategies. Specifically, **RL4VLM** applies standard **token-level** PPO to VLMs, while **VAGEN** employs a **bi-level** advantage estimation strategy to stabilize online training.
>
> The experimental results for **ERA, RL4VLM, and VAGEN** are directly cited from the era paper[2] (October 2025), where these methods were evaluated under a unified setting on EmbodiedBench.
>
> A critical observation is that these baselines utilize heavy data augmentation during their SFT stage, whereas our original method relied on a smaller, unaugmented dataset. This created a performance disparity in the base SFT models. **To ensure an fair comparison of the optimization methods** (Ours RL vs. Online RL), we re-trained our SFT backbone using the same augmented dataset used by these baselines. This places all methods on the same starting line. We then applied our RFT with offline dataset on top of this enhanced SFT model.
>
> **2. Experimental Results**
>
> The results on the **EB-ALFRED** benchmark are summarized below:
>
> | **Method**          | Ours (with augmented SFT)  | ERA  | **WAP** | **RL4VLM** | **VAGEN** |
> | ------------------- | ------------------------------ | -------- | ------- | ---------- | --------- |
> | **EB-ALFRED (Avg)** | **70.88**                      | 65.2     | 62.7    | 51.2       | 52.8      |
>
> Our method (70.88%) outperforms the strong Online RL baseline ERA (65.2%) and significantly surpasses RL4VLM and VAGEN. This validates that our **dense, prefix-based offline reward** effectively mitigates the sparse reward problem inherent in online embodied planning, providing a more stable learning signal than sparse success/failure feedback.
>
> In summary, our approach not only avoids the high computational cost of online interaction but also delivers superior performance compared to standard Online RL methods in this domain.
>
>
>
> [1] From Multimodal LLMs to Generalist Embodied Agents: Methods and Lessons", CVPR 2025.
>
> [2] Era: Transforming vlms into embodied agents via embodied prior learning and online reinforcement learning." arXiv:2510.12693
>
> [3] World-aware Planning Narratives Enhance Large Vision-Language Model Planner." arXiv:2506.21230
>
> [4] Fine-tuning large vision-language models as decision-making agents via reinforcement learning, Neurips 2024
>
> [5] VAGEN: Reinforcing world model reasoning for multi-turn vlm agents. arXiv:2510.16907

---

> ### Author Response · Authors · 2025-11-26
> **Response (3/5)**
>
> # W3: Precision of Data Filtering Thresholds
>
> > W3:  The description of the data filtering process in Lines 190-193 lacks the necessary precision. The manuscript uses ambiguous terms such as 'too few' and 'too many' without providing formal, quantitative definitions. To ensure the method is reproducible, the authors must specify the exact numerical thresholds or statistical criteria used to implement this filtering step.
>
> We thank the reviewer for pointing out the ambiguity in our description of the data filtering process. We agree that precise definitions are essential for reproducibility.
>
> The terms "too few" and "too many" refer to the success rate of the generated responses within a group for a given prompt. Specifically, we filter out a prompt group if the proportion of responses achieving a perfect reward (accuracy = 1.0) is:
>
> - **Lower than 0.1:** Indicating the task is too difficult for the current policy to learn from effectively (too few positive signals).
> - **Higher than 0.9:** Indicating the task is already mastered, providing negligible gradient information (too many perfect signals).
>
> These specific numerical thresholds (accuracy_lower_bound = 0.1 and accuracy_upper_bound = 0.9) were documented in Appendix Table 4 due to space constraints in the main text.
>
> To address this concern, we will revise Section 2.2 (Lines 190-193) in the final manuscript to explicitly state these values (0.1 and 0.9), ensuring the method is formally defined and fully reproducible without needing to consult the appendix.
>
>
>
>
>
> # Q1: Robustness to Sub-Optimal Trajectories
>
> > Q1: The proposed approach relies on expert trajectories for reward computation. How robust is this method if, as is often the case in real-world scenarios, expert data is expensive or unavailable, and only sub-optimal trajectories can be obtained?
>
> We thank the reviewer for raising this practical concern regarding the availability and quality of expert data. We address this from both theoretical and empirical perspectives.
>
> **1.Theoretical Perspective: Optimality is Not Strict**
>
> In embodied planning, "expert" trajectories are rarely uniquely optimal; a task can often be solved via multiple valid action sequences. Our theoretical analysis in Section 2.3.2 (Proposition 1) anticipates this. We prove that the policy optimization remains directionally correct and bounded relative to the true environment objective, even if the reference trajectory is just one of many valid solutions. The reward signal $R_{accuracy}$ primarily encourages alignment with successful behavior patterns rather than strict memorization of a single "golden" path.
>
> **2.Empirical Evidence: Perturbation Study**
>
> To empirically quantify robustness against sub-optimal data, we conducted a controlled experiment where we intentionally degraded the quality of our RFT training dataset. We introduced noise into the expert trajectories using two perturbation strategies:
>
> - **Insertion of Irrelevant Actions:** Randomly injecting non-functional actions into the sequence.
> - **Redundant Loops:** Inserting closed-loop navigation paths (e.g., moving away and returning) to create inefficient trajectories.
>
> We trained our model using datasets with **10%** and **20%** of the trajectories perturbed, respectively. The results on the EB-ALFRED benchmark (using our augmented SFT backbone for RFT) are shown below:
>
> | **RFT Data Quality**           | **Avg**   | **Base** | **Com** | **Cplx** | **Visual** | **Spatial** |
> | ------------------------------ | --------- | -------- | ------- | -------- | ---------- | ----------- |
> | **0% Perturbation (Standard)** | **70.88** | 76.4     | 64      | 82       | 72         | 60          |
> | **10% Perturbation**           | **68.88** | 76.4     | 64      | 76       | 68         | 60          |
> | **20% Perturbation**           | 62.50     | 74.5     | 52      | 76       | 62         | 48          |
>
> - **High Resilience at 10% Noise:** Even with 10% of the training data corrupted by sub-optimal noise, the model maintains an impressive average success rate of **68.88%**. Notably, this still outperforms the state-of-the-art proprietary model **Claude-3.5-Sonnet (66.4%)**.
> - **Graceful Degradation at 20% Noise:** At 20% perturbation, while performance drops to 62.5%, the model remains highly competitive, significantly outperforming the base GPT-4o model (56.8%).
>
> This confirms that our method does not require "perfect" expert data to succeed and can robustly extract planning policies even from datasets containing a significant portion of sub-optimal or noisy trajectories.

---

> ### Author Response · Authors · 2025-11-26
> **Response (4/5)**
>
> # Q2：Reward definition and the meaning of n,k
>
> > Q2: Could you clarify the meaning of the terms k+1 and n+1 as they appear in equation (3)?
>
> In our reward function $R_{accuracy} = \frac{n(n+1)}{k(k+1)}$:
>
> - **$k$** represents the fixed length of the **ground-truth expert trajectory** for the given task.
> - **$n$** represents the length of the **longest correct prefix** matched by the model's generated rollout
>
> We selected this quadratic form over linear alternatives ($n/k$) based on our ablation study (Section 3.3), which showed it yields superior performance. Theoretically, the quadratic scaling assigns increasing marginal utility to later steps in a sequence.
>
>
>
> # Q3：Handling Out-of-Distribution (OOD) Trajectories
>
> > Q3：How are out-of-distribution issues addressed in the reward computation? Specifically, when the policy diverges from the expert policy, do you assume it always leads to suboptimal actions?
>
> We address out-of-distribution (OOD) actions through a strict prefix-matching strategy, where any deviation from the expert trajectory halts reward accumulation. This acts as a regularizer to prevent "reward hacking" in the absence of online verification. Crucially, unlike SFT, our **on-policy** optimization allows the model to learn from its own distribution—including self-generated negative examples—which provides stronger resistance to forgetting and superior generalization when facing OOD scenarios.

---

> ### Author Response · Authors · 2025-11-28
> **Response (5/5)**
>
> # Q4：Variance and Confidence Intervals
> > Q4：The results in Table 1 are presented without confidence intervals (CIs). Given that reinforcement learning methods are known to have high variance, CIs are essential for a statistically robust evaluation and for making reliable comparisons between approaches. Please add confidence intervals (or standard deviations) for all results and specify the number of runs (e.g., random seeds) used to compute these statistics.
>
> We acknowledge the importance of statistical robustness in reinforcement learning. It is worth noting that the evaluation environment (EmbodiedBench) is relatively deterministic, resulting in naturally low variance compared to highly stochastic environments.
>
> To quantitatively validate this, we conducted **5 independent training runs** using different random seeds. We report the detailed raw data for each run below, followed by the summary statistics.
>
> **Table 1: Detailed Raw Results across 5 Independent Runs (%)**
>
> | **Run ID** | **Base** | **Com** | **Cplx** | **Visual** | **Spatial** | **Avg** |
> | ---------- | -------- | ------- | -------- | ---------- | ----------- | ------- |
> | **Run 1**  | 76       | 64      | 82       | 72         | 60          | 70.8    |
> | **Run 2**  | 78       | 60      | 84       | 74         | 60          | 71.2    |
> | **Run 3**  | 78       | 64      | 82       | 72         | 62          | 71.6    |
> | **Run 4**  | 78       | 64      | 80       | 71         | 58          | 70.2    |
> | **Run 5**  | 76       | 64      | 82       | 72         | 64          | 71.6    |
>
> **Table 2: Statistical Summary (Mean $\pm$ Standard Deviation)**
>
> The low standard deviations across all metrics confirm that our method's performance improvements are stable and statistically significant.
>
> | **Method** | **Avg**              | **Base**        | **Com**         | **Cplx**        | **Visual**      | **Spatial**     |
> | ---------- | -------------------- | --------------- | --------------- | --------------- | --------------- | --------------- |
> | **Ours **  | **71.08 $\pm$ 0.53** | 77.2 $\pm$ 0.98 | 63.2 $\pm$ 1.60 | 82.0 $\pm$ 1.26 | 72.2 $\pm$ 0.98 | 60.8 $\pm$ 2.04 |

---

### Official Review · Reviewer_YqGx · 2025-11-08

**Soundness:** 2
**Presentation:** 3
**Contribution:** 2
**Rating:** 2
**Confidence:** 4

**Summary:**

This paper proposes an on-policy reinforcement fine-tuning (RFT) framework for multi-step embodied task planning, aiming to close the gap between VLMs that perform well on static perception tasks and those required to act coherently in dynamic environments.

The method introduces an offline reward mechanism that compares model-generated trajectories with expert demonstrations, thereby avoiding costly simulator interaction and sparse environmental feedback. Rewards are computed via a quadratic prefix-based function measuring how long the predicted action sequence matches the expert trajectory (Eq. 3), combined with a small format-regularization term (Eq. 4). Optimization is performed through GRPO.

The authors claim theoretical guarantees showing that policies optimized under offline rewards remain bounded with respect to true online-reward policies. Experiments on Embench demonstrate improvements over both open-source and closed-source baselines. On EB-ALFRED (in-domain), their model reaches 49.2% success rate, outperforming Qwen2.5-VL-72B (40.8%) and LLaMA-3.2-90B (35.2 %); on EB-Habitat (out-of-domain) it achieves 22.4%, outperforming similarly-sized models.

Ablation studies test reward allocation and computation strategies, showing the prefix-based reward performs better than embedding-similarity metrics and that combining accuracy (1.0) + format (0.5) yields the best results. Overall, the work claims to enable scalable, simulation-free RFT for embodied reasoning.

**Strengths:**

- Clear motivation and strong empirical performance: The problem of costly online reward collection in embodied environments is well-motivated (in the real world at least), and experiments on Embench show tangible gains over large-scale baselines.
- The reward function is sound and simple. The optimal policy under this reward also proves close to the optimal policy under a sparse environment reward. Empirical results also show the effectiveness of this design.
- Comprehensive benchmarking and ablations: a good collection of open source and closed baselines models of difference sizes and reward variants are compared, offering a broad empirical picture.

**Weaknesses:**

### Unclear definition of the underlying MDP / “on-policy” claim
Section 2.1 formulates a POMDP with observations o_t and actions a_t, but Section 2.2 states that training avoids simulation. If the agent never steps an environment, there is no transition function P(s’|s,a) and thus no genuine new observation after an action. The process effectively samples full trajectories from the model and scores them offline against stored expert data, an imitation-style static optimization, not an interactive MDP. The paper should clarify what “on-policy” means here and explicitly state that the “environment” is frozen in the dataset. As written, the theoretical framing in Sec. 2.3.2 over-extends beyond what is actually optimized.

---

### Reward definition lacks justification and rigor.
In Eq. 3 the reward assumes the predicted and expert action sequences have equal length $k$, yet this is never guaranteed: model outputs can terminate early or differ in length. It is unclear how mismatched sequences are handled. The quadratic term $n(n+1)/k(k+1)$ appears ad hoc: the "$+1$" factor and quadratic scaling are not theoretically motivated beyond "smoothness".

---

### Limited discussion of computational efficiency.
The paper repeatedly claims that online reinforcement fine-tuning in embodied simulators is "prohibitively expensive" and motivates the offline-reward design on this basis. However, no quantitative evidence is provided: there are no comparisons of training time, GPU hours, or memory/FLOP usage between the proposed offline RFT and a standard online-interaction setup.

Moreover, modern embodied simulators are parallelizable: dozens or hundreds of episodes can run concurrently on a single node using GPU-based rendering. For these environments, the cost of stepping simulations is often negligible compared with large-model forward/backward passes. Thus, the argument that simulator interaction is the main computational bottleneck is not well-substantiated.

**Questions:**

- The authors should clarify the exact decision process underlying their “on-policy” RFT framework: if no simulation is performed, what defines the MDP’s transition dynamics and how are new observations obtained after each predicted action?
- Relatedly, the term “on-policy” needs to be precisely defined: does it merely refer to sampling from the current model, or does the data distribution evolve in any way to reflect policy behavior?
- The reward formulation also requires further justification: it assumes equal-length action sequences between model and expert trajectories, yet the paper does not explain how mismatched lengths are handled, nor why the quadratic prefix reward was chosen over simpler alternatives.
- Finally, the claimed computational advantage of avoiding simulator interaction should be supported with quantitative evidence. It would be helpful to specify whether it is possible to use parallelized simulation for training within EmbodiedBench, and to report actual training or rollout costs. Clarifying these points would make the technical contributions and efficiency claims much more convincing.

---

> ### Author Response · Authors · 2025-11-26
> **Response (1/3)**
>
> # W1 & Q1/Q2: Clarification on MDP Formulation and "On-Policy" Definition
>
> We thank the reviewer for the sharp observation regarding our problem formulation. We acknowledge that the distinction between the **training** decision process and the **evaluation** environment was not sufficiently emphasized in Section 2.1. Below, we clarify the exact nature of our MDP, the training/inference gap, and our definition of "on-policy."
>
> **1. Clarification of the Underlying Decision Process (Q1)**
>
> > Q1：The authors should clarify the exact decision process underlying their “on-policy” RFT framework: if no simulation is performed, what defines the MDP’s transition dynamics and how are new observations obtained after each predicted action?
>
> The reviewer is correct that our training phase does not involve a dynamic simulator transition function P(s'|s,a). Instead, we decouple the training and evaluation processes as follows:
>
> * **Training Phase (Single-Step MDP / Contextual Bandit):**
>   As described in the "Offline Expert Trajectory Construction" (L268-271) and similar to standard RLHF (e.g., PPO/GRPO for LLMs), we formulate training as a **single-step MDP** (or Contextual Bandit).
>   * **State ($s$):** A composite input containing the task instruction $g$, the history of actions $a_{0:n-1}$, and the visual observation $o_n$ recorded in the expert dataset.
>   * **Action ($a$):** The model predicts the **entire remaining sequence** of actions $\hat{a}_{n:} = \{a_n, \dots, a_k\}$.
>   * **Transition & Observation:** Since the model outputs the full future plan in one turn relative to the provided state, the episode ends immediately after the prediction. No new observation $o_{n+1}$ is needed from a simulator because the reward is computed by comparing the predicted plan against the offline expert ground truth.
>   * **Reward:** The reward is computed offline based on the correctness of the generated sequence relative to the expert trajectory.
>
> * **Evaluation Phase (Multi-Turn Dynamic MDP):**
>   During evaluation (Section 3.1), we strictly follow the **EmbodiedBench** protocol. This is a fully interactive, multi-turn MDP where the agent predicts actions step-by-step (or in chunks), executes them in the simulator (AI2-THOR/Habitat), and receives actual state transitions and new visual observations.
>
>
>
> **2. Definition of "On-Policy" (Q2)**
>
> > Q2：Relatedly, the term “on-policy” needs to be precisely defined: does it merely refer to sampling from the current model, or does the data distribution evolve in any way to reflect policy behavior?
>
> In our framework, **"on-policy"** is defined as **sampling trajectories directly from the current policy $\pi_\theta$ for optimization**, rather than using a fixed data buffer or teaher-forcing method.
>
> * **Specifically:** We sample $a \sim \pi_\theta(\cdot|s)$. The model generates its own rollouts (plans) based on the input state. These rollouts are then scored, and the policy is updated to increase the probability of high-reward rollouts generated by the *current* model.
> * **Why this distinction matters:** This fundamentally differs from Supervised Fine-Tuning (SFT). As analyzed in Section 2.3.1 and corroborated by recent theoretical work [1-2] , **on-policy sampling** is critical for generalization and preventing forgetting. Unlike SFT, which imitates fixed data, our approach optimizes the *current* policy distribution, allowing the model to learn from self-generated negative examples and thereby achieving stronger robustness.
>
>
>
> [1] RL's Razor: Why Online Reinforcement Learning Forgets Less. arXiv:2509.04259.
>
> [2] Retaining by Doing: The Role of On-Policy Data in Mitigating Forgetting. arXiv:2510.18874
>
>
>
>
>
> **3. Validity of Theoretical Analysis (W1)**
>
> > W1： The process effectively samples full trajectories from the model and scores them offline against stored expert data, an imitation-style static optimization, not an interactive MDP. As written, the theoretical framing in Sec. 2.3.2 over-extends beyond what is actually optimized.
>
> We respectfully suggest that our theoretical framing in Section 2.3.2 remains valid under this Single-Step MDP formulation. The proofs regarding the bound between the optimal policies rely on the divergence of the **reward landscapes** over the outcome space $\mathcal{Y}$, not on the specific transition dynamics.
>
> Specifically, Proposition 1 proves that optimizing the policy under our dense, offline expert reward ($r_{exp}$) yields a policy that is boundedly close to the one optimized under the sparse environment reward ($r_{env}$), provided the reward functions are defined on the same output space. This holds true whether that output space represents a single token or a full trajectory generated in a single turn.
>
> We will revise Section 2.1 in the final version to explicitly state the Single-Step formulation during training to prevent confusion with the interactive multi-step evaluation setting.

---

> > ### Author Response · Authors · 2025-11-26
> > **response (2/3)**
> >
> > # W2 & Q3: Clarification on Reward Definition and Justifying Quadratic Reward Design
> >
> > We appreciate the reviewer identifying the ambiguity in our reward formulation. We would like to clarify the definitions of $n$ and $k$, and acknowledge a notational error in the text.
> >
> > **1.Definition of Variables $n$ and $k$**
> >
> > In our reward function $R_{accuracy} = \frac{n(n+1)}{k(k+1)}$:
> >
> > - **$k$** represents the fixed length of the **ground-truth expert trajectory** for the given task.
> >
> > - **$n$** represents the length of the **longest correct prefix** matched by the model's generated rollout2.
> >
> >
> >
> > **2.Handling Mismatched Sequence Lengths**
> >
> > > W2&Q3：In Eq. 3 the reward assumes the predicted and expert action sequences have equal length , yet this is never guaranteed: model outputs can terminate early or differ in length. It is unclear how mismatched sequences are handled
> >
> > Critically, the length of the model's predicted sequence (rollout) does not need to equal the expert trajectory length $k$.
> >
> > - If the model generates a sequence shorter than $k$ (early termination), $n$ is simply the number of correct steps before termination.
> > - If the model generates a sequence longer than $k$, $n$ is capped at $k$ (assuming the prefix matches up to that point), and subsequent tokens do not contribute to the accuracy reward.
> > - Consequently, neither our reward calculation nor our theoretical analysis requires the assumption of equal lengths. The denominator relies solely on the expert trajectory length $k$
> >
> >
> >
> >  **3.Why the quadratic prefix reward was chosen over simpler alternatives.**
> >
> > > W2&Q3： The paper does not explain why the quadratic prefix reward was chosen over simpler alternatives.
> >
> > + **Empirical Evidence:** As detailed in our ablation study (Section 3.3 and Figure 3a), we explicitly compared our quadratic term against simpler alternatives, including a **linear prefix reward** ($n/k$) and embedding-based similarity rewards. The results demonstrate that the quadratic formulation consistently yields superior performance.
> >
> >   | **Reward Formulation**      | **EB-ALFRED (Seen) Success Rate (%)** | **EB-Habitat (Unseen) Success Rate (%)** |
> >   | --------------------------- | ------------------------------------- | ---------------------------------------- |
> >   | **Ours (Quadratic Prefix)** | **49.2**                              | **22.4**                                 |
> >   | Linear-Rank ($n/k$)         | 42.0                                  | 15.6                                     |
> >   | Embed-Cosine                | 40.8                                  | 19.6                                     |
> >   | Embed-OT                    | 46.8                                  | 16.0                                     |
> >
> > + **Theoretical Motivation:** In multi-step embodied planning, later steps are conditional on the success of earlier ones and are therefore harder to reach. A simple linear reward ($n/k$) assigns equal marginal value to the first step and the last step. In contrast, the quadratic term ($\propto n^2$) assigns increasing marginal utility to extending the correct prefix length. This acts as a "soft curriculum," encouraging the model not just to memorize easy starting steps (which would yield decent linear rewards) but to strive for long-horizon coherence, as the reward gain is highest when completing the full sequence. This aligns with our goal of "long-horizon-aware shaping".
> >
> >
> >
> > **4.Correction of Notation Error**
> >
> > We acknowledge that the notation in Section 2.2, "Let $\hat{a}=\{a_1, ..., a_k\}$ be the predicted sequence" is imprecise. It incorrectly implies that the predicted sequence length is fixed to $k$. We will correct this in the final revision to denote the predicted sequence as $\hat{a}=\{a_1, ..., a_m\}$, where $m$ is the arbitrary length of the model's generation.

---

> ### Author Response · Authors · 2025-11-26
> **response (3/3)**
>
> # W3 & Q4: Computational Efficiency and Engineering Constraints
>
> > W3&Q4: The paper repeatedly claims that online reinforcement fine-tuning in embodied simulators is "prohibitively expensive" and motivates the offline-reward design on this basis. However, no quantitative evidence is provided. It would be helpful to specify whether it is possible to use parallelized simulation for training within EmbodiedBench, and to report actual training or rollout costs.
>
> We thank the reviewer for raising this important point regarding computational efficiency. While we agree that modern simulators are theoretically capable of high-parallelism, we would like to clarify that the "cost" in VLM reinforcement learning stems not just from rendering time, but from the **systemic complexity of integrating interactive environments with Large Language Model training loops**.
>
> **1. Engineering Complexity and VLM-RL Integration**
>
> While simulators like AI2-THOR are parallelizable, integrating them into a high-throughput training loop for VLMs (e.g., 7B+ parameters) presents significant engineering hurdles. Unlike standard RL where policies are small (MLPs), VLMs require massive GPU memory. Efficient, large-scale frameworks that seamlessly couple Embodied Simulators with VLM training are still in their infancy. Mature frameworks for this specific integration, such as SimpleVLA-RL[1] , RLinf[2], ERA[3]  have only begun to emerge very recently (September 2025), which is subsequent to the completion of our work.
>
> Our "Offline Reward" approach bypasses these engineering bottlenecks entirely. By decoupling the environment from the optimization loop, we can leverage standard, highly optimized LLM training frameworks (like OpenRLHF/DeepSpeed) without modification, treating the VLM training as a standard text-generation task with a computed reward.
>
>
>
> [1] "Simplevla-rl: Scaling vla training via reinforcement learning." arXiv:2509.09674.
>
> [2] "RLinf-VLA: A Unified and Efficient Framework for VLA+ RL Training."  arXiv:2510.06710.
>
> [3] Era: Transforming vlms into embodied agents via embodied prior learning and online reinforcement learning." arXiv:2510.12693
>
>
>
> **2. Quantitative Comparison of Computational Costs**
>
> To address the request for quantitative evidence, we conducted a direct comparison between our **Offline RFT** and a standard **Online RL** setup (based on the ERA framework). We measured the training time per update step, total convergence time, and final performance on the EB-ALFRED benchmark. All experiments were conducted on 8x H200 GPUs, though our method is deployable on a minimum configuration of 4x 80G VRAM.
>
> **Table: Comparison of Training Efficiency and Performance**
>
> | **Metric**                  | **Ours (Offline RFT)**                                       | **Online RL (Baseline)**                                     | **Comparison**        |
> | --------------------------- | ------------------------------------------------------------ | ------------------------------------------------------------ | --------------------- |
> | **Time per Update Step**    | **~2 min**                                                   | 12 - 15 min                                                  | **~6x - 7.5x Faster** |
> | **Total Training Time**     | **8 - 10 hours**                                             | 36 - 40 hours                                                | **~4x Faster**        |
> | **Final Performance (Avg)** | **70.88%**                                                   | 65.2%                                                        | **+5.68% Higher**     |
> | **Environment Setup**       | **Standard VLM+RL**  (e.g., OpenRLHF, VeRL).  Simple to deploy. | **Complex Integration**  (Simulator + Headless Rendering/Xorg + VLM-RL Framework). High engineering barrier. | **Plug-and-Play**     |
>
> As shown in the table, the Online RL method suffers from significant latency due to multi-step simulator rendering, environment resets, and the synchronization overhead between the VLM inference and the physics engine (taking 12-15 mins per step). In contrast, our method decouples the environment, treating the task as efficient text generation with fast string-matching rewards, reducing the step time to just **2 minutes**. Crucially, this efficiency does not come at the cost of performance. Our method achieves a higher success rate (**70.88%**) in significantly less time (**8-10 hours** vs. 36-40 hours).
>
>
>
> **3.Real-world Cost**
>
> Finally, it is crucial to note that while simulators *can* be parallelized, **real-world online training** is prohibitively expensive and often infeasible due to safety risks and high reset costs. Our method, by leveraging offline data for policy optimization, avoids these physical risks entirely, offering a safer and more scalable path for real-world RL.

---

### Author Response · Authors · 2025-11-28
**Summary of Our Rebuttal and Look forward to Discussions**

We sincerely thank all reviewers and Area Chairs for their time and effort in evaluating our work. We appreciate the constructive feedback, which has helped us clarify our contributions and strengthen our empirical validation.

We are encouraged that reviewers recognized our work as well-motivated (Reviewers YqGX, 9WSH) in addressing the critical bottlenecks of costly simulation and sparse rewards in embodied AI. Reviewers consistently highlighted the soundness of our reward design (YqGX, yvuV), its theoretical justification (TFDL), our comprehensive benchmarking (YqGX) and the practicality of our scalable approach (Reviewer TFDL) as significant strengths.

During the rebuttal phase, we focused on addressing the specific concerns raised regarding technical definitions, computational costs, and robustness:

- **Clarification of Technical Concepts:** We have provided precise definitions and justifications for our MDP formulation, the "on-policy" nature of our training loop, and the reward computation mechanism to resolve conceptual ambiguities.
- **Quantitative Efficiency Analysis:** We added detailed experiments comparing computational costs, demonstrating that our method is **4-5x faster** than standard Online RL fine-tuning and requires significantly lower engineering complexity.
- **Comparison with Recent SOTA:** We benchmarked our approach against very recent methods (e.g., ERA[1], WAP[2]) that appeared in the last few months. Under identical experimental settings, our method achieves a new **State-of-the-Art average success rate of 70.88%** on EB-ALFRED.
- **Robustness Verification:** We conducted perturbation studies to test resilience against sub-optimal data. Even with **10% and 20% noisy data** injected into the expert trajectories, our model’s performance dropped by only **2% and 7.5%** respectively, proving its strong robustness.

Overall, we believe our work proposes a novel and promising paradigm for embodied AI: leveraging the generalization benefits of on-policy RFT while bypassing the prohibitive costs of online interaction via offline rewards. Specifically, the core paradigm of **utilizing offline expert trajectories to derive dense reward signals for on-policy reinforcement fine-tuning** strongly aligns with several emerging trends [3-6] in the broader community (these works were released subsequent to our submission), validating the shift towards scalable policy optimization.

We will incorporate all additional experiments, theoretical clarifications, and baselines into the final version of the manuscript. Given the limited time remaining in the discussion period, we are happy to engage in further discussion to address any remaining questions.

Best regards,

The Authors



[1] Era: Transforming vlms into embodied agents via embodied prior learning and online reinforcement learning." arXiv:2510.12693

[2] World-aware Planning Narratives Enhance Large Vision-Language Model Planner." arXiv:2506.21230

[3] Supervised Reinforcement Learning: From Expert Trajectories to Step-wise Reasoning, arXiv:2510.25992

[4] "On-Policy Distillation", Thinking Machines Lab, https://thinkingmachines.ai/blog/on-policy-distillation

[5]  Retaining by Doing: The Role of On-Policy Data in Mitigating Forgetting. arXiv:2510.18874

[6] SRPO: Self-Referential Policy Optimization for Vision-Language-Action Models  arXiv:2511.15605

---

### Note · Authors · 2026-01-05

I have read and agree with the venue's withdrawal policy on behalf of myself and my co-authors.